# Impairment of a distinct cancer-associated fibroblast population limits tumour growth and metastasis

Ute Jungwirth [1,2], Antoinette van Weverwijk [1,3], Rachel J. Evans [1], Liam Jenkins[1], David Vicente [1], John Alexander [1], Qiong Gao [1,4], Syed Haider [1], Marjan Iravani [1] & Clare M. Isacke [1✉]

Profiling studies have revealed considerable phenotypic heterogeneity in cancer-associated fibroblasts (CAFs) present within the tumour microenvironment, however, functional characterisation of different CAF subsets is hampered by the lack of specific markers defining these populations. Here we show that genetic deletion of the Endo180 (*MRC2*) receptor, predominantly expressed by a population of matrix-remodelling CAFs, profoundly limits tumour growth and metastasis; effects that can be recapitulated in 3D co-culture assays. This impairment results from a CAF-intrinsic contractility defect and reduced CAF viability, which coupled with the lack of phenotype in the normal mouse, demonstrates that upregulated Endo180 expression by a specific, potentially targetable CAF subset is required to generate a supportive tumour microenvironment. Further, characterisation of a tumour subline selected via serial in vivo passage for its ability to overcome these stromal defects provides important insight into, how tumour cells adapt to a non-activated stroma in the early stages of metastatic colonisation.

[1] The Breast Cancer Now Toby Robins Research Centre, The Institute of Cancer Research, London, UK. [2] Department of Pharmacy and Pharmacology, Centre for Therapeutic Innovation, University of Bath, Bath, UK. [3] Present address: Division of Tumor Biology & Immunology, The Netherlands Cancer Institute, Amsterdam, The Netherlands. [4] Present address: Strategic Initiatives, Data Science, The Institute of Cancer Research, Sutton, UK. ✉email: clare.isacke@icr.ac.uk

There is extensive functional evidence implicating cancer-associated fibroblasts (CAFs) in tumour progression, via their ability to deposit and remodel the extracellular matrix (ECM), to secrete pro-tumourigenic factors and by modulating the immune compartment[1–5]. However, there is also evidence that stromal fibroblasts can play a role in restraining tumour growth, for example by acting as a desmoplastic barrier to tumour cell invasion and by the recruitment of anti-tumour immune cells[6]. Consideration of how to limit the pro-tumourigenic functions of CAFs, whilst retaining their anti-tumourigenic role, has been hampered by a lack of understanding of the heterogeneity of CAFs within solid tumours, and the paucity of specific markers to identify and functionally analyse different CAF subpopulations. Progress in addressing these shortcomings has come from a number of studies reporting single cell sequencing of CAFs or analysis of CAF subsets, which has revealed the diversity of fibroblast populations and provided an important clues as to their origin and functional properties[7–14].

In this study, we explore the functional role of a CAF receptor, Endo180 (also known as uPARAP) encoded by the *MRC2* gene. The 180 kDa Endo180 receptor comprises an N-terminal cysteine-rich domain, a fibronectin type II (FNII) domain that has been shown to bind collagens[15–20], 8 C-type lectin-like domains (CTLDs), of which only CTLD2 displays $Ca^{2+}$-dependent binding of sugars[21], a single pass transmembrane domain and a cytoplasmic domain that interacts with components of the clathrin-mediated internalisation machinery to drive constitutive recycling between the plasma membrane and intracellular endosomes[22,23]. Although Endo180 is expressed by some sarcomas[24], glioblastomas[25] and metaplastic breast cancers[26], in most solid tumours of epithelial origin, expression of Endo180 is reported to be predominantly restricted to CAFs with little or no expression by the tumour cells[27–30]. Functionally, Endo180 can mediate endosomal uptake of collagens for lysosomal degradation[18,20] and has been demonstrated to promote cell migration[15,16], via its ability to generate Rho-ROCK-MLC2-mediated contractility[31].

Here, we report that Endo180 expression is associated with the subset of CAFs characterised by their expression of matrix components and matrix-modifying enzymes and that whole-body genetic deletion of Endo180, although having little phenotype in normal mice, severely limits the growth and metastatic colonisation of inoculated wildtype syngeneic tumour cells. Mechanistic studies reveal that failure to upregulate Endo180 expression in the tumour microenvironment results in CAFs with impaired contractility and reduced viability, and that this in turn can drive an altered pattern of tumour evolution resulting in the selection of tumour cells with enhanced intrinsic matrix remodelling properties.

## Results

### Endo180 expression by cancer-associated fibroblasts (CAFs).
Consistent with the published literature[27–30], immunohistochemical staining of breast (Fig. 1a) and other solid tumours (Supplementary Fig. 1a) reveals high levels of Endo180 protein on fibroblasts within the tumour stroma. The cell type specificity of this receptor was confirmed by RT-qPCR analysis of cell populations sorted from dissociated mouse mammary tumours, demonstrating that *Mrc2* (Endo180) expression is restricted to the PDGFRα-positive CAF population with no detectable expression on macrophages or other immune cells, endothelial cells, or tumour cells (Fig. 1b and Supplementary Fig. 1b). An equivalent cell-type distribution is seen in other tumour types. For example, Endo180 (*MRC2*) expression is restricted to the CAF population sorted from human colorectal tumours[32], and represents one of the 30 genes whose expression is significantly higher in fibroblasts

versus all other sorted populations (Supplementary Fig. 1c). Similarly, single cell RNA-Seq analysis of mouse melanoma primary tumours and draining lymph nodes[33], reveals *Mrc2* expression to be exclusively restricted to CAFs with no expression on other cell types, including lymph node fibroblasts (Supplementary Fig. 1d). Importantly, expression in CAFs represents upregulated levels of Endo180 in the tumour stroma as illustrated by significantly higher *MRC2* expression, alongside increased expression of two other fibroblast activation markers, α-smooth muscle actin (αSMA; *ACTA2*) and fibroblast activation protein (*FAP*), in a gene expression dataset of microdissected breast cancer stroma compared to adjacent normal tissue stroma[34] (Fig. 1c). A similar increase in expression is seen in CAFs isolated from mouse mammary tumours compared to normal mouse mammary gland fibroblasts (Fig. 1d), and can be recapitulated in vitro in co-culture of MRC5 normal human fibroblasts with human breast cancer cells or mouse mammary carcinoma cells (Fig. 1e, left panel), with equivalent results seen in published datasets profiling Wi38 or HFF1 human fibroblasts co-cultured with breast cancer cells (Fig. 1e, right panels). These data indicate that increased *MRC2* expression in tumours does not solely reflect a proportionate increase in the number of fibroblasts in the tumour stroma. Equivalent levels of *MRC2* expression are seen in CAFs isolated from oestrogen receptor positive (ER+), human epidermal growth factor receptor 2-positive (HER2+) or triple negative breast cancers (TNBCs) (Fig. 1f) indicating that the upregulated Endo180 expression in CAFs is not breast cancer subtype restricted. Consistent with these data, when taking all breast cancer subtypes together *MRC2* expression strongly correlates with a fibroblast TGFβ response signature (F-TBRS) (Fig. 1g), that serves as marker of fibroblast activation in the tumour stroma and is prognostic of poor outcome in colorectal cancers[32,35].

As well as inter-tumour heterogeneity within the stromal compartments of solid cancers, there is increasing evidence of intra-tumoural stromal heterogeneity, including CAF heterogeneity[7–14,33]. Single cell sequencing of fibroblasts isolated from the mammary tumours in MMTV-PyMT transgenic mice[7] identified three fibroblast subsets defined as vascular CAFs, matrix CAFs and developmental CAFs plus a group of actively cycling vascular CAFs termed cycling CAFs. Endo180 is most strongly expressed by matrix CAFs (mCAFs; Fig. 1h), the subset characterised by expression of ECM and ECM-related genes and predicted to originate from activated resident fibroblasts. Similarly, Endo180 is most strongly expressed by the S2 CAF subpopulation, characterised by expression of ECM components, in the melanoma single cell RNA-Seq dataset[33] (Supplementary Fig. 1d) whilst FAP and αSMA display a broader distribution including expression on lymph node fibroblasts, together indicating that Endo180 expression is associated with a discrete population of CAFs in solid tumours. Comparison of normal skin fibroblasts and CAFs in the mouse melanoma dataset[33] and of normal tissue fibroblasts and tumour-associated fibroblast clusters in a human lung cancer single cell RNA-Seq dataset[36] confirms upregulated *Mrc2/MRC2* expression in subsets of CAFs (Supplementary Fig. 1e, f).

### Impaired tumour growth and metastasis in Endo180$^{-/-}$ mice.
To investigate the functional consequence of the upregulated Endo180 expression by CAFs, mice with a whole-body targeted deletion in Endo180 (*Mrc2*)[15] were backcrossed onto a BALB/c background and inoculated orthotopically with syngeneic 4T1 mouse mammary carcinoma cells. In wildtype mice, 4T1 tumours grow at the expected rate (Fig. 2a) and recruit Endo180-positive cells into the tumour stroma (Fig. 2b). Post-mortem examination of the lungs reveals readily detectable spontaneous macrometastatic disease in five out of six mice. By contrast, primary tumours

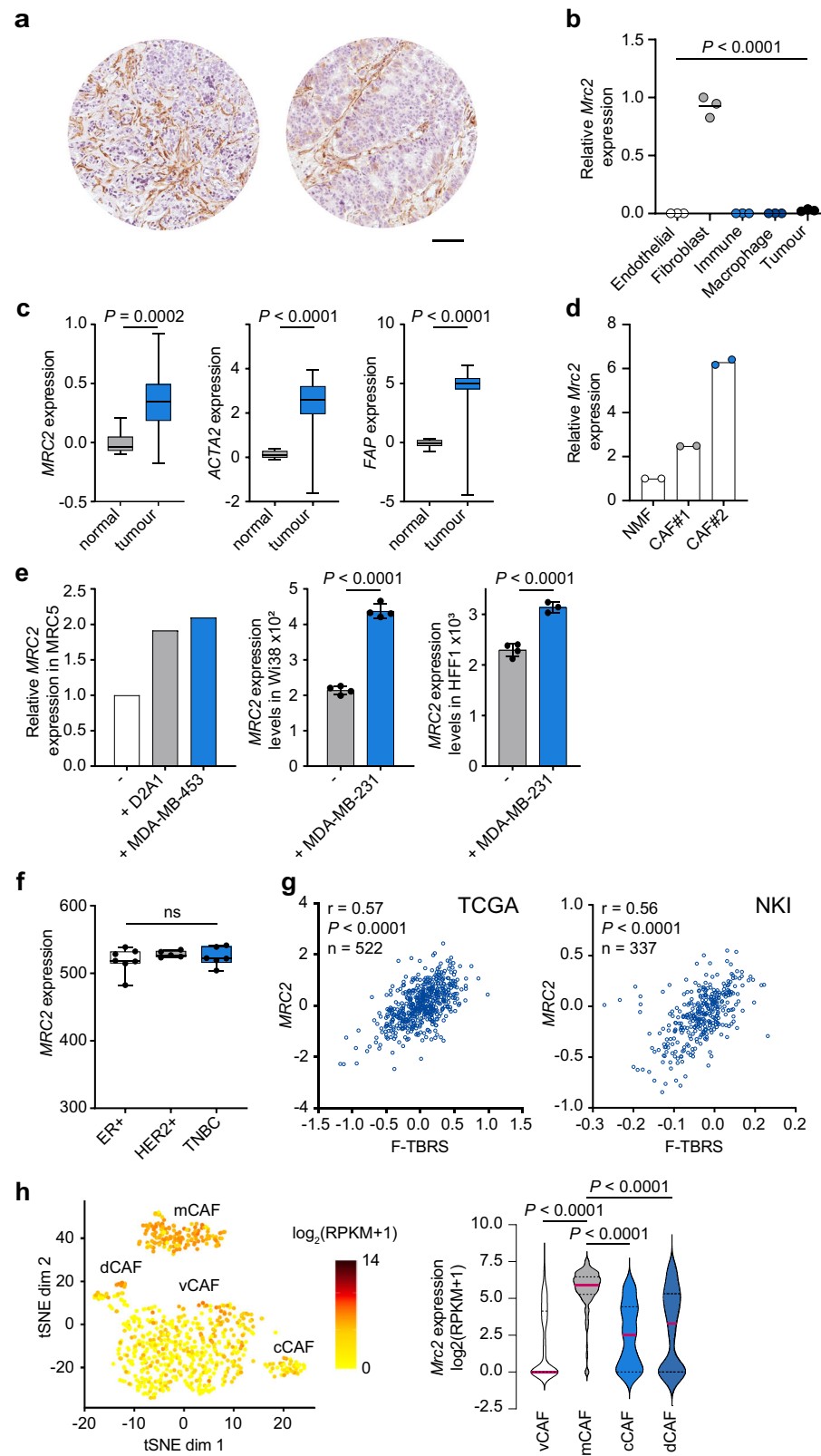

only develop in three out of six Endo180$^{-/-}$ mice and those mice which do develop tumours show a delay in tumour growth, a significant reduction in spontaneous metastasis and, as expected, no detectable Endo180 staining (Fig. 2a, b). To confirm these findings, the experiment was repeated with the D2A1 mouse mammary carcinoma cell line. Tumour take was only observed in four out of nine Endo180$^{-/-}$ mice and, again, the tumours which

do develop in the Endo180$^{-/-}$ mice grow more slowly than in wildtype mice such that at the termination of the experiment, there is a significant difference in tumour volume (Fig. 2c) and tumour weight (Supplementary Fig. 2a). D2A1 cells are poorly metastatic and therefore an impairment in metastasis could not be ascertained, however, we recently derived the metastatic D2A1-m2 subline[37]. After orthotopic inoculation, the D2A1-m2

**Fig. 1 Endo180 is associated with fibroblast activation. a** Immunohistochemical staining of Endo180 (mAb 39.10) of a tumour microarray of invasive human breast cancers[26]. Illustrative images, scale bar, 0.1 mm. **b** Endo180 (*Mrc2*) expression in endothelial cells, fibroblasts, immune cells, macrophages and tumour cells isolated from mouse D2A1-m2 tumours (data from three individual mice; one-way ANOVA with Tukey's multiple comparison test). **c** Boxplots of Endo180 (*MRC2*), *ACTA2* and *FAP* expression in microdissected breast cancer stroma ($n = 53$ samples) and adjacent normal tissue stroma ($n = 6$ samples). Box plots shows median and 25th–75th quartiles, whiskers show minimum and maximum. (GSE901[34], two-sided Mann–Whitney *U*-test). **d** Relative Endo180 (*Mrc2*) expression in cultured normal mouse mammary gland fibroblasts and two independent CAF cultures isolated from 4T1 tumours ($n = 2$). **e** Left panel, relative Endo180 (*MRC2*) expression in human MRC5 fibroblasts cultured alone or co-cultured with D2A1 or MDA-MB-453 tumour cells. Middle and right panels, Endo180 (*MRC2*) expression in Wi38 and HFF1 human fibroblasts, respectively, cultured alone or co-cultured with MDA-MB-231 tumour cells (GSE41678[72]; Wi38 panel and HFFF1 alone, $n = 4$; HFFF1 co-culture, $n = 3$; mean values ± SEM, two-sided *t*-test). **f** Endo180 (*MRC2*) expression in fibroblasts isolated from different breast cancer subtypes. Box plots shows median and 25th–75th quartiles, whiskers show minimum and maximum. (GSE37614[71]; ER+, $n = 7$; $n = 5$ HER2+, $n = 5$; TNBC, $n = 6$; one-way ANOVA with Tukey's multiple comparison test, all non-significant (ns) $P > 0.05$). **g** Correlation of Endo180 (*MRC2*) gene expression with a fibroblast TGFβ response signature (F-TBRS)[32] in breast cancers from the TCGA (Pearson correlation, $r = 0.57$, $P < 0.0001$ $n = 522$) and the NKI295[70] (Pearson correlation, $r = 0.56$, $P < 0.0001$, $n = 337$) datasets. **h** Endo180 (*Mrc2*) gene expression in the published single cell sequencing dataset of CAFs isolated from MMTV-PyMT tumours[7]. Shown is *Mrc2* expression log$_2$(RPKM + 1) in individual cells (left panel, tSNE plot) and in the vascular (vCAF, $n = 485$), matrix (mCAF, $n = 144$), cycling (cCAF, $n = 43$) and development (dCAF, $n = 44$) subpopulations (Violin plots, red bars indicate median expression levels, dotted lines indicate upper and lower quartiles, Kruskal–Wallis test, Dunn's correction). Source data are provided as a Source Data file.

primary tumours grow equally well in both wildtype and Endo180$^{-/-}$ mice, but there remains a significant defect in their ability to spontaneously metastasise in the Endo180$^{-/-}$ mice (Fig. 2d and Supplementary Fig. 2b) confirming that all three cell lines examined show impaired tumour progression, but that the defects manifested vary between the lines.

To address directly whether Endo180$^{-/-}$ mice show an impairment in their ability to support metastatic growth, tumour cells were inoculated via the tail vein into recipient mice, which results predominantly in single cells disseminating to the lungs. BALB/c mice inoculated with 4T1 cells, the related but less aggressive 4T07 cells, D2A1 cells, and D2A1-m2 cells all develop large macrometastatic lesions in the lungs of wildtype mice but are severely impaired in their growth in Endo180$^{-/-}$ mice as monitored by IVIS imaging for luciferase tagged cells, ex vivo lung weight or quantification of tumour burden in histological sections (Fig. 2e–h and Supplementary Fig. 2c–e). Importantly, equivalent results are obtained following intravenous inoculation of C57BL/6 Endo180$^{-/-}$ mice with the syngeneic E0771 mouse mammary carcinoma line (Fig. 2i and Supplementary Fig. 2f) demonstrating that the defect in lung colonisation is not restricted to the BALB/c background. Finally, we also investigated whether this effect was evident in other organs by performing intrasplenic inoculation of 4T1 cells. Whereas all wildtype mice developed liver metastases, only 1 out of 8 Endo180$^{-/-}$ mice showed evidence of macrometastatic disease (Fig. 2j and Supplementary Fig. 2g).

Together these data suggest that upregulated expression of Endo180 in matrix CAFs promotes efficient tumour progression in vivo. To test further this assertion, we took two approaches. First, D2A1 tumour cells were injected orthotopically into BALB/c mice alone or admixed with shNTC or shEndo180 GFP-positive CAFs. As reported recently in a CAF consensus statement[5], fibroblasts recruited from the host stroma outgrow co-injected CAFs making the analysis of long-term phenotypes hard to test. Co-injection of shNTC CAFs, but not shEndo180 CAFs, results in a more consistent tumour growth, as monitored by the variation in final tumour size, but the final tumour volumes are not significantly different (Supplementary Fig. 3a, b). Consistent with the consensus statement[5], all tumours contained αSMA-positive cells but no co-injected GFP-CAFs are detectable at the end of the experiment (Supplementary Fig. 3c, d). Second, we examined tumour colonisation in tissues that lack stromal fibroblasts. Endo180 expression is not detected in the normal brain apart from weak expression in some perivascular cells[25] and similarly there is lack of expression of classic fibroblast markers, such as

FSP1 and αSMA (Supplementary Fig. 3e). Following direct intracranial inoculation of 4T1 tumour cells there is no significant difference in brain tumour burden in wildtype and Endo180$^{-/-}$ mice (Supplementary Fig. 3f). Similarly, there is no significant difference in brain metastasis following intracardiac inoculation of 4T1 cells (Supplementary Fig. 3g).

**Reduced contractility and viability of CAFs lacking Endo180.** To investigate the mechanism by which an Endo180-deficient stroma impairs tumour progression, fibroblasts were co-seeded with tumour cells into low adherence U-bottom plates to generate 3D co-culture spheroids. Growth of D2A1 tumour cell spheroids, as monitored by spheroid size and CellTiter-Glo viability, is significantly enhanced when admixed with mouse CAFs, immortalised normal mouse mammary gland fibroblasts (NF#1) or NIH-3T3 fibroblasts (Fig. 3a, b). By contrast, admixing with fibroblasts that had been transfected with Endo180 siRNAs (Supplementary Fig. 4a) severely impairs fibroblast-mediated enhanced spheroid growth. Equivalent results are obtained with D2A1 tumour cells admixed with CAFs at different ratios, D2A1 cells admixed with mouse CAFs transduced with two independent shRNAs targeting Endo180, and 4T1 tumour cells admixed with CAFs or normal fibroblasts (Supplementary Figs. 4b, c and 5a–c). However, although conditioned medium from wildtype CAFs can promote the growth of D2A1 tumour spheroids, conditioned medium from siEndo180 transfected CAFs is equally as effective (Fig. 3c). Similarly, there is no difference in D2A1 2D colony formation in the presence of mock, siNTC and siEndo180-transfected CAF conditioned medium (Fig. 3d), indicating that Endo180 expression does not regulate the production of soluble tumour growth promoting components. Consequently, we addressed next whether the ECM produced by Endo180 wildtype and Endo180-deficient CAFs differentially modulated tumour cell growth. shRNA transduced CAFs were cultured alone or with conditioned medium from 4T1 tumour cells to generate CAF-derived ECM. Fibronectin staining of the decellularised matrices allows an examination of the ECM deposition and fibre alignment using the TWOMBLI macro in FIJI[38]. Compared to wildtype CAFs, Endo180-deficient CAFs cultured with or without 4T1 conditioned medium produce a reduced density matrix (Fig. 3e). Treatment of CAFs with 4T1-conditioned medium results in a more anisotropic matrix, indicated by the reduced number of normalised end and branching points, however, there is no difference in anisotropy in Endo180-deficient compared to the Endo180 wildtype CAF-derived matrices. Moreover, while 4T1 and D2A1 tumour cells grow

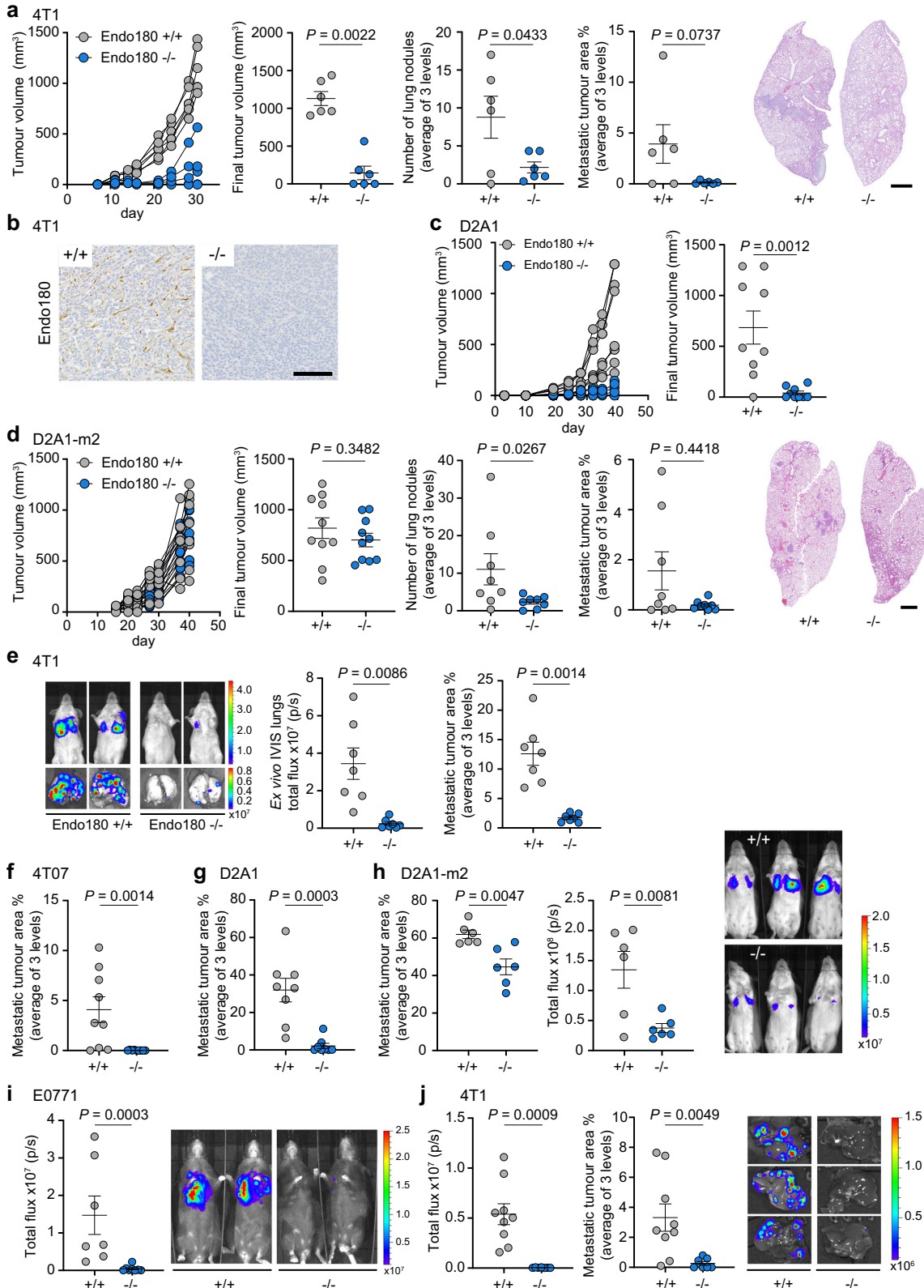

better on a CAF-derived ECM compared to a 3T3 fibroblast-derived ECM, there is again no difference in tumour cell proliferation (Fig. 3f and Supplementary Fig. 5d) or tumour cell colony formation (Fig. 3g) on matrices derived from Endo180 wildtype or Endo180-deficient CAFs.

As the inability of Endo180-deficient CAFs to support tumour growth could not be recapitulated by CAF conditioned medium

or CAF-derived ECM, we next addressed CAF intrinsic properties. When plated onto collagen-coated soft (2 kPa) hydrogels, wildtype CAFs contract their cytoskeleton and round up, whereas Endo180-deficient cells remain spread resulting in a significantly increased single cell area and decreased circularity (Fig. 4a). This effect is not specific to a collagen substratum or a soft substratum as equivalent results are observed with fibronectin-coated

**Fig. 2 Endo180 promotes tumour growth, spontaneous and experimental metastasis.** All graphs show mean values ± SEM, $t$- and Mann–Whitney $U$-tests are two-sided. **a–d** Tumour cells were injected into the 4th mammary fat pad of BALB/c Endo180[+/+] or Endo180[−/−] mice. Spontaneous lung metastasis was quantified as mean nodule number and mean % metastatic area in three lung sections. Representative lung sections (scale bar, 1 mm). **a** $1 \times 10^4$ 4T1-Luc-RFP cells inoculated ($n = 6$ per group). Tumour growth in individual mice (no tumours were detected in three Endo180[−/−] mice), final tumour volume (Mann–Whitney $U$-test), quantification of spontaneous lung metastasis ($t$-tests). **b** 4T1 tumours stained for Endo180 (scale bar, 100 μm). **c** $5 \times 10^4$ D2A1 cells inoculated ($n = 9$ per group). No tumours detected in four Endo180[−/−] and one Endo180[+/+] mouse. Final tumour volume (Mann–Whitney $U$-test). **d** $5 \times 10^4$ D2A1-m2-Luc cells inoculated ($n = 10$ per group). Final tumour volume ($t$-test). Mice with intraperitoneal tumour growth were excluded from metastatic analysis. Quantification of lung nodules ($t$-test) and % metastatic area (Mann–Whitney $U$-test). **e** $1 \times 10^5$ 4T1-Luc cells injected intravenously into BALB/c mice ($n = 7$ per group). Quantification of day 12 ex vivo lung IVIS imaging ($t$-test) and % metastatic area ($t$-test). **f** $2.5 \times 10^5$ 4T07 cells injected intravenously into BALB/c mice ($n = 9$ per group). Day 24 quantification of metastatic area (Mann–Whitney $U$-test). **g** $4 \times 10^5$ D2A1 cells injected intravenously into BALB/c mice ($n = 8$ per group). Day 23 quantification of metastatic area (Mann–Whitney $U$-test). **h** $4 \times 10^5$ D2A1-m2-Luc cells injected intravenously into BALB/c mice ($n = 6$ per group). Day 14 in vivo IVIS images, quantification of ex vivo lung IVIS imaging ($t$-test) and metastatic area ($t$-test). **i** $4 \times 10^5$ E0771-Luc cells injected intravenously into C57BL/6 mice (Endo180[+/+], $n = 7$; End180[−/−], $n = 8$). Day 17 quantification of in vivo lung IVIS imaging (Mann–Whitney $U$-test). **j** $1 \times 10^5$ 4T1-Luc cells injected intrasplenically into BALB/c mice (Endo180[+/+], $n = 9$; Endo180[−/−], $n = 7$). Day 13 quantification of ex vivo liver IVIS imaging ($t$-test) and % metastatic area (three sections; Mann–Whitney $U$-test). Additional data are provided in Supplementary Fig. 2. Source data are provided as a Source Data file.

hydrogels (Fig. 4b), and when fibroblasts are plated onto collagen or fibronectin-coated stiff (50 kPa) hydrogels or glass coverslips (Supplementary Fig. 6a, b). Moreover, when embedded into a collagen matrix, Endo180-deficient CAFs show a striking inability to efficiently contract the collagen gels (Fig. 4c) or to invade into a 3D matrix (Fig. 4d and Supplementary Fig. 6c). To determine whether the contractility defect could be overcome, 3T3 fibroblasts, which show reduced levels of contractility when embedded in a collagen matrix, were treated with TGFβ, a key driver of fibroblast activation both in vitro and in vivo[32]. TGFβ treatment dramatically promotes collagen gel contraction by wildtype 3T3 fibroblasts, but has little impact on the ability of Endo180-deficient fibroblasts to contract the gels (Fig. 4e). Despite these clear differences in fibroblasts contractility, there are no robust differences in the levels of key actomyosin regulators, phosphorylated myosin light chain (pMLC) and phosphorylated myosin phosphatase targeting subunit 1 (pMYPT) between wildtype and Endo180-deficient fibroblasts (Fig. 4f), however, there is a distinct difference in pMLC subcellular distribution. In wildtype fibroblasts treated with or without TGFβ, pMLC is associated with the entire length of the actin stress fibres, whereas in Endo180-deficient fibroblasts pMLC distribution is more concentrated within the body of the cell (Fig. 4g and Supplementary Fig. 7a). Moreover, despite a substantial increase in expression of αSMA (*Acta2*) and *Fap*, classic markers of fibroblast activation, in CAFs compared to 3T3 fibroblasts, there is no significant difference in expression of these markers following siRNA or shRNA-mediated Endo180 down-regulation (Fig. 5a–c). Similarly, Endo180-deficient fibroblasts show no difference in the expression of *Col1a1*, *Col1a2*, or MLC (*Myl9*). Finally, in agreement with previous reports[39], CAFs show an enhanced level of YAP/TAZ translocated into the nucleus compared to 3T3 fibroblasts, however, downregulation of Endo180 has no impact on YAP/TAZ nuclear localisation (Fig. 5d).

The finding that Endo180-deficient CAFs in culture retain markers of fibroblast activation led us to test the hypothesis that the CAF contractility defect has a cell intrinsic effect on Endo180[−/−] fibroblast viability in the tumour stroma. Endo180-deficient fibroblasts plated into non-adherent conditions in low serum, but not 10% serum, have an increased level of apoptosis (Fig. 6a). Similarly, while wildtype CAFs plated alone in U-bottom plates form stable 3D aggregates that show a modest increase in size over time, Endo180-deficient CAF 3D aggregates deteriorate such that they are significantly smaller at day 10 (Fig. 6b). More importantly, in 3D spheroid co-culture assays significantly fewer Endo180-deficient fibroblasts can be recovered from dissociated spheroids

(Fig. 6c). Consistent with these findings, 4T1, D2A1, and D2A1-m2 primary tumours in Endo180[−/−] mice have a significant deficit in intra-tumoural αSMA-positive cells (Fig. 7a), and a reduction in intra-tumoural fibrillar collagen deposition as detected by Masson's trichrome staining (Fig. 7b). By contrast, no differences are detectable either in the infiltration of F4/80 macrophages (Fig. 7c) or in the vascular architecture (Fig. 7d), indicating that there is not a global disruption of the tumour stroma.

**Tumour adaptability to stromal deficiency.** Tumours can evolve to overcome therapeutic insult or the lack of a supportive microenvironment[40] and consequently it was important to investigate how tumours might adapt to overcome the micro-environmental block associated with loss of Endo180. Previously we described the selection of the D2A1-m2 metastatic subline from the non-metastatic D2A1 mouse mammary carcinoma cell line by repeated in vivo passage in wildtype BALB/c mice[37]. Here we used a similar approach to generate a D2A1 subline, D2A1-m12, by serial in vivo passage in Endo180-deficient mice (see "Methods" section). Compared to the parental D2A1 cells, the selected D2A1-m12 subline rapidly forms macrometastatic lesions in the Endo180[−/−] BALB/c mice following intravenous inoculation (Fig. 8a) and, in contrast to the parental and other mouse mammary carcinoma lines tested (Fig. 2e–i), gives rise to an equivalent metastatic burden when inoculated intravenously into Endo180[−/−] and wildtype mice (Fig. 8b). Moreover, in a spontaneous metastasis experiment, D2A1-m12 primary tumours show an increased growth rate in Endo180[−/−] mice, albeit with no significant difference in tumour weight at necropsy, and no difference in the metastatic burden in the lungs (Supplementary Fig. 8a).

In in vitro assays, admixing D2A1 or D2A1-m12 cells with CAFs significantly promoted 3D co-culture spheroid growth (Fig. 8c). However, as reported in Fig. 3a, whereas Endo180-deficient CAFs are impaired in their ability to promote the growth of D2A1 3D co-culture spheroids there is no significant difference in D2A1-m12 spheroid growth, when admixed with wildtype or Endo180-deficient fibroblasts (Fig. 8c). Further, whereas D2A1 cells admixed with either wildtype or Endo180-deficient fibroblasts are unable to effectively mediate collagen-gel contraction, gel contraction is more pronounced with the D2A1-m12 cells even when admixed with Endo180-deficient fibroblasts (Supplementary Fig. 8b). Together these data support the notion that tumour cell selection can generate sublines that can overcome, at least in part, the requirement for fibroblast support both in vitro and in vivo. To address the mechanism by which this may occur, D2A1 and D2A1-m12 cells were subject to whole

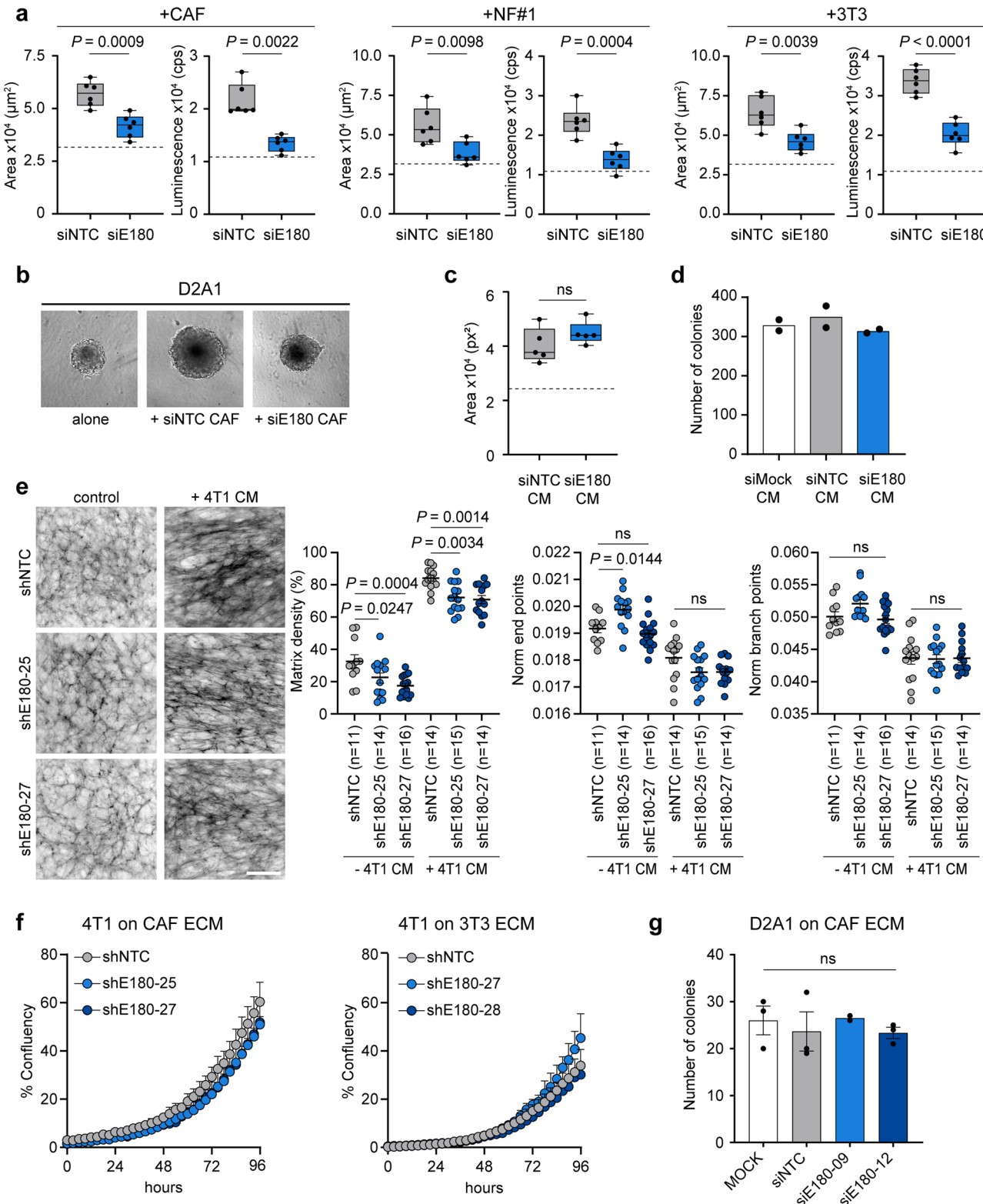

exome sequencing and transcriptional profiling. Exome sequencing reveals that, when compared to a reference BALB/c genome, the D2A1 and D2A1-m12 cell lines display a similar copy number aberration profile (Supplementary Fig. 8c) as expected given that the D2A1-m12 subline is derived from the D2A1 parental cells. Comparison of their mutation profiles reveals a similar mutation burden in the D2A1 and D2A1-m12 cell lines (Supplementary

Fig. 8d). Although there is a significant overlap of shared mutations (Supplementary Fig. 8e), both cell lines also carry distinct mutations, consistent with the parental D2A1 containing a heterogenous mix of subclones and the D2A1-m12 subline being derived from a subset of these during the in vivo passage. RNA-Seq analysis of the D2A1 and D2A1-m12 cells revealed 1203 significantly differentially expressed genes (DEGs;

**Fig. 3 Spheroid co-cultures recapitulate in vivo Endo180 impact. a, b** Fibroblast-tumour cell co-culture spheroid assays. **a** 300 D2A1 cells alone or in combination with 600 mouse CAFs, NF#1 or 3T3 fibroblasts transfected with non-targeting control (NTC) or Endo180 (E180) targeting siRNA oligonucleotides were seeded into U-bottom low adherence plates and cultured for 8 days. Spheroid growth was assessed by spheroid area and CellTiter-Glo. Box plots shows median and 25th–75th quartiles, whiskers show minimum and maximum. Dotted lines show mean spheroid size/CellTiter-Glo value of tumour cells alone ($n = 6$ spheroids per condition; mean values ± SEM; panel 2, two-sided Mann–Whitney $U$-test; other panels, two-sided $t$-test). **b** Representative images of D2A1 spheroids alone or co-cultured with siNTC or siE180-transfected CAFs (scale bar, 200 μm). **c** D2A1 spheroids cultured for 8 days in serum-free conditioned medium (CM) collected from siRNA-transfected CAFs supplemented with 2% FBS. Box plots show median and 25th–75th quartiles, whiskers show minimum and maximum. Dotted line shows mean spheroid size in control DMEM plus 2% FBS ($n = 5$ spheroids per condition; mean values ± SEM, two-sided Mann–Whitney $U$-test). **d** D2A1 colony formation assay in the presence of CM from mock, siRNA-transfected CAFs and supplemented with 2% FBS ($n = 2$). **e** CAFs transduced with shNTC or two independent shRNAs targeting Endo180 cultured with or without CM from 4T1 cells supplemented with 2.5% FBS. Cultures were stained for fibronectin. Left panel, representative images of fibronectin fibres (scale bar, 100 μm). Right panels, quantification of matrix density, normalised endpoints and normalised branch points ($n = 11$–16 fields of view per condition as indicated; mean values ± SEM, two-way ANOVA, Tukey post-hoc test). **f** 4T1 cell growth, measured by confluency, on ECM derived from CAFs or 3T3 fibroblasts transduced with shNTC or two independent shRNAs targeting Endo180 ($n = 3$ wells per condition; mean values ± SEM, two-way ANOVA, non-significant in all comparisons). **g** D2A1 colony formation assay on ECM derived from mock transfected CAFs or CAFs transfected with siNTC or two independent siRNAs targeting Endo180 (MOCK siNTC, siE180-12, $n = 3$; mean values ± SEM, one-way ANOVA; siE180-09, $n = 2$). Non-significant (ns) $P$ values > 0.05. Source data are provided as a Source Data file.

Supplementary Fig. 9a, b), including a 5.27 and 2.69 logFC increase in αSMA (*Acta2*) and Endo180 (*Mrc2*) expression, respectively, in D2A1-m12 cells. This upregulated expression was validated by RT-qPCR analysis of cultured tumour cells and tumours cells isolated from in vivo tumours, although it is important to note that expression of both *Acta2* and *Mrc2* in the D2A1-m12 cells is still substantially lower than expression levels in CAF (Supplementary Fig. 9c). We performed fast Gene Set Enrichment Analysis (fGSEA) of the RNA-Seq data using GSKB pathways supplemented with the mouse core matrisome and matrisome-associated pathways from the Matrisome Project[41], due to the lack of well annotated ECM components in most gene ontology categories (Fig. 8d; Supplementary Table 1). Notably, within the nine significantly enriched pathways in the D2A1-m12 cells (FDR adjusted $P$ value < 0.1) are core matrisome collagens, core matrisome glycoproteins, matrisome-associated glycoproteins and integrin signalling (Fig. 8e and Supplementary Fig. 9d; see Supplementary Table 2 for list of top 50 upregulated matrisome genes in D2A1-m12 cells). Finally, we determined whether the altered matrisomal gene expression was retained in vivo. D2A1 and D2A1-m12 tumours were dissociated and tumour cells isolated by FACS. RT-qPCR analysis validated the increased expression of selected matrisomal collagens and glycoproteins identified in the D2A1-m12 RNA-Seq data and, conversely, validated the downregulated expression of *Bgn* and *Ltpb2* (Fig. 8f) These findings suggest that the enhanced tumourigenesis of the D2A1-m12 subline in Endo180$^{-/-}$ mice results from tumour cells acquiring an enhanced ability to remodel and respond to the ECM, when CAF viability is impaired.

## Discussion

CAFs are one of the most abundant cell types within the tumour stroma, however it is now clear from the extensive transcriptional profiling of CAF subsets and, more recently, single cell sequencing studies that the term CAF represents a phenotypically and functionally diverse mix of cells. Traditionally, CAFs have been proposed to promote tumourigenesis via their ability to deposit and remodel the ECM, and by their ability to interact both with tumour cells and other stromal cell types via the secretion of growth factors, chemokines, and cytokines. However, both normal stromal fibroblasts and CAFs have also been accredited with tumour suppressive properties, and it remains unclear how tumour cells orchestrate their interactions with the micro-environment to overcome fibroblast/CAF mediated suppression and take advantage of the pro-tumourigenic CAF functions. One

explanation for observed CAF diversity in tumours is that they might have different origins[5,6,42,43]. Certainly, CAFs can arise from the activation of resident stromal fibroblasts, and these may well represent the bulk of the CAFs associated with early lesions. In addition, perivascular cells from a wide variety of tissues have been demonstrated to exhibit phenotypic and functional properties associated with mesenchymal stem cells[44,45]. In this respect it is of interest that in their CAF single cell sequencing study, despite excluding cells expressing the pericyte marker NG2, Bartoschek and colleagues defined a large population of fibroblasts as vascular CAFs which they propose originate from a pool of perivascular cells[7]. The second most prominent population identified were matrix CAFs, characterised by the expression of a wide variety of ECM and ECM-related genes that are proposed to originate from the activation of resident fibroblasts. Here, we have focussed on the fibroblast receptor Endo180 which is predominantly expressed by the matrix CAF subset. Interestingly, a comparative analysis on mice with a genetic deletion in a vascular CAF receptor endosialin (*Cd248*) revealed a strikingly different phenotype compared to the Endo180 knockout mice. Whereas the reduced metastasis observed in the endosialin$^{-/-}$ mice was due to a defect to tumour cell dissemination[46], the Endo180$^{-/-}$ mice display a defect in tumour growth, and metastatic colonisation supporting the contention of different functional roles for different fibroblast subsets.

Although Endo180 is expressed at low level on stromal fibroblasts in normal adult tissues it is notable that non-tumour bearing whole-body Endo180 knockout mice have no notable phenotypic abnormalities and are born at the expected Mendelian frequency[15,16,47]. However, a combined knockout of both Endo180 and the membrane type 1-matrix metalloproteinase (MT1-MMP), a major component of extracellular collagen degradation, resulted in accelerated postnatal lethality and an impairment in bone growth[47]. These data, in agreement with our current and other studies[28], suggest that loss of Endo180 in normal tissues can be compensated for, but that under conditions where matrix remodelling is compromised such as by loss of MT1-MMP or in the altered conditions of the tumour micro-environment, a requirement for functional Endo180 is revealed. This hypothesis is supported by a previous study reporting that crossing an Endo180$^{-/-}$ mouse with the MMTV-PyMT mouse reduced primary tumour burden[28]. However, the effects on metastatic colonisation were not investigated and little insight was provided as to the underlying mechanisms. In particular, these previous studies have focussed on the role of Endo180 as an ECM remodelling receptor via its ability to internalise collagen

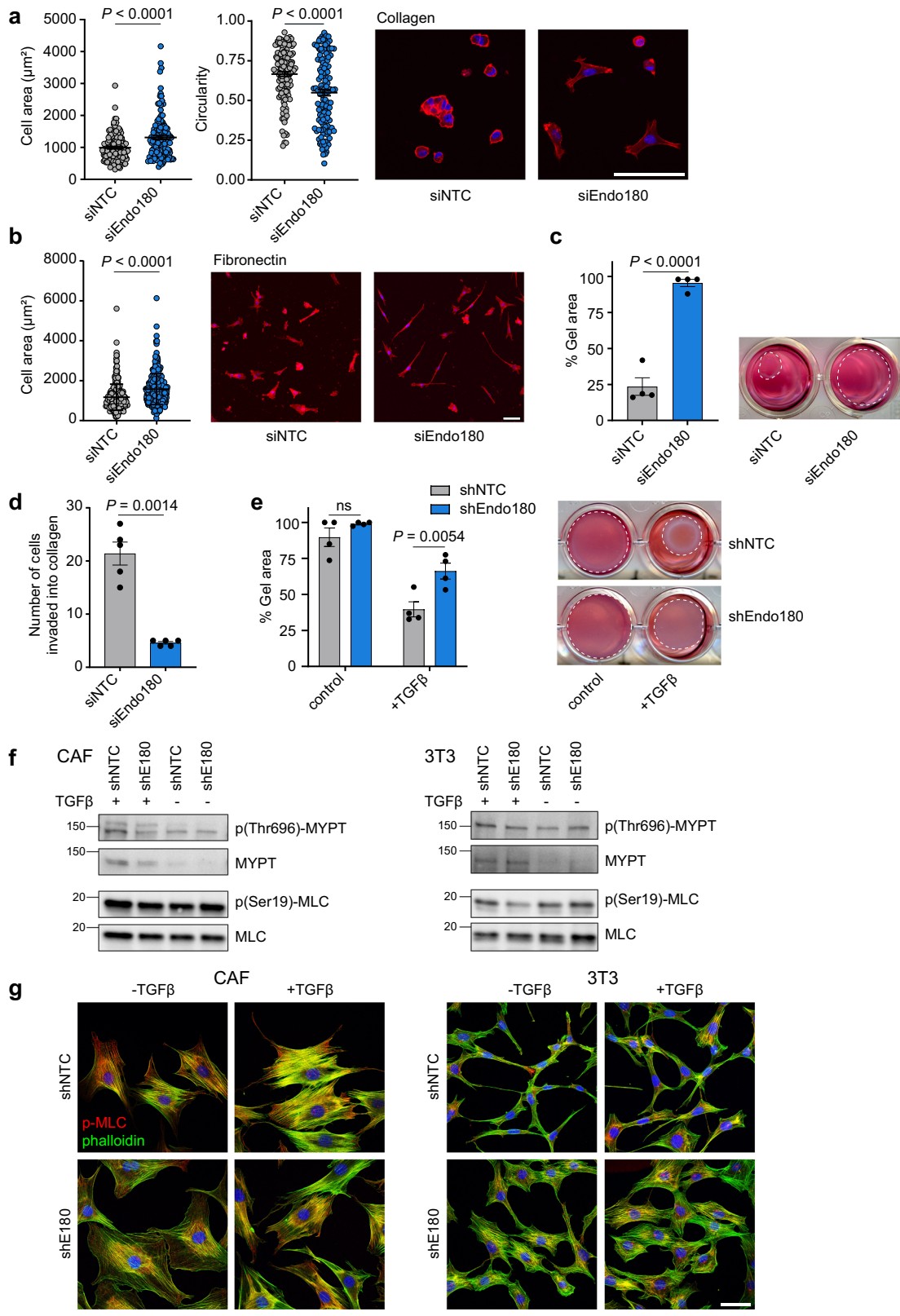

for intracellular degradation. Of note, in contrast to our findings of decreased intra-tumoural collagen content in the 4T1 and D2A1-m2 tumours (Fig. 7), in the PyMT tumours, genetic deletion of Endo180 was reported to result in an increased accumulation of collagens[28]. This difference may reflect that in the PyMT tumours Endo180-positive cells and collagen deposition is predominantly surrounding the tumour nests[28], an area where fibroblasts accumulate but are less activated compared to the tumour infiltrating CAFs[48].

Although there is no question that Endo180 can function as a collagen receptor[15–18,20,28,49], we demonstrate here that Endo180-deficient fibroblasts show an equivalent impaired

**Fig. 4 Endo180-deficient fibroblasts show reduced contractility. a, b** Cell spreading (area) and cell circularity of siNTC and siEndo180 CAFs cultured on soft (2 kPa) hydrogels coated with **a** collagen ($n = 133$ cells; mean values ± SEM, two-sided Mann-Whitney $U$) or **b** fibronectin ($n = 300$ cells; mean values ± SEM, Mann–Whitney $U$-test). Cells were stained with DAPI and Alexa555-phalloidin. Right panels show representative images (scale bar, 100 μm). **c** Collagen contraction assay. shNTC or shEndo180 transfected CAFs were embedded into rat tail collagen (2 mg mL$^{-1}$). Data show % gel area ($n = 4$, mean values ±SEM, two-sided $t$-test) after 24 h. Right panel, representative images. **d** Number of siNTC or siEndo180 CAFs invading from fibroblast aggregates embedded in a collagen matrix (2 mg mL$^{-1}$) after 24 h ($n = 5$ wells; mean values per well ± SEM, two-sided $t$-test). **e** shNTC or shEndo180 3T3 fibroblasts were embedded in collagen I gels in DMEM plus 2% FBS with/without TGFβ (5 ng mL$^{-1}$). Data show % gel area after 48 h ($n = 4$, mean values ± SEM, two-way ANOVA). **f, g** Cultured shNTC and shE180 transduced CAFs or 3T3s were starved overnight and then stimulated with or without TGFβ (5 ng mL$^{-1}$) for 1 h. **f** Western blots of p(Thr696)-MYPT, MYPT, p(Ser19)-MLC and MLC. Molecular size markers are in kDa. Ponceau loading controls are shown in Supplementary Fig. 6d. **g** Confocal images of p(Ser19)-MLC (red), phalloidin (green), and DAPI stained cells (scale bar, 50 μm). Images without phalloidin staining are shown in Supplementary Fig. 7. Source data are provided as a Source Data file.

contractility when plated onto either collagen or fibronectin-coated hydrogels (Fig. 4 and Supplementary Fig. 6). Similarly, we have previously reported that the enhanced cell migration and contractile signalling associated with ectopic Endo180 expression in Endo180-negative cells is not collagen dependent[31], but that it is dependent on Endo180 internalisation/recycling as the pro-migratory phenotype is not observed following expression of the collagen binding, but internalisation-defective, Endo180(Ala$^{1468}$/Ala$^{1469}$) cytoplasmic domain mutant[22,31,50]. These data point to a CAF intrinsic defect in the Endo180$^{-/-}$ tumour stroma and indeed we demonstrate here that Endo180 depletion is associated with increased fibroblast apoptosis when plated in 3D low serum culture and decreased fibroblast viability in 3D tumour-spheroid co-culture. Importantly, this is echoed by a reduced fibroblast content within the tumour stroma of Endo180$^{-/-}$ mice (Fig. 7). Interestingly, Wagenaar–Miller and colleagues reported that the Endo180 single knockout mice showed no difference in the number of apoptotic cells in the developing bones compared to wildtype mice. However, there was significantly increased apoptosis and a corresponding decrease in cell proliferation in the double Endo180/MT1-MMP knockout mice compared to either single knockout[47]. Further, upregulation of Endo180 expression has been shown for pathological conditions such as liver fibrosis induced by CCl$_4$[51], lung fibrosis induced by bleomycin[52], and during wound healing[53,54]. During wound healing, myofibroblasts are characterised by increased expression of αSMA[55] but show an inverse correlation between proliferation and ECM deposition[56]. At the end of the normal wound healing process[57] or during the resolution of fibrosis[58], myofibroblasts undergo cell death. However, evasion of apoptosis can lead to fibrosis via persistent myofibroblast activation and contraction-induced latent TGFβ activation[57,59]. Furthermore, it has been shown that a reduction in mechanical stiffness in the matrix leads to an increase in fibroblast apoptosis[60,61]. Hence, we suggest that a parallel situation may occur in tumours and that Endo180-deficient fibroblasts, which have reduced contractility and responsiveness to TGFβ (Fig. 4), will be cleared from the stromal compartment, whilst in wildtype CAFs Endo180 expression serves to maintain cell viability in pathological conditions.

Finally, we addressed how tumours might respond to a defective microenvironment by selecting for a tumour subline, D2A1-m12, by serial in vivo passage in Endo180$^{-/-}$ mice. Compared to the parental D2A1 cells, D2A1-m12 cells were characterised by a striking upregulation of ECM and ECM-associated genes, as well as by increased αSMA (Acta2) and Endo180 (Mrc2) expression. Together, these data suggest that D2A1-m12 subline is selected for a subclone(s) enriched for specific properties required to overcome the stromal deficit in the Endo180$^{-/-}$ mice; providing further evidence for the critical role played by the tumour microenvironment in driving tumour evolution. This is supported by a recent study showing that tumour and stromal cells together shape the metastatic niche with

the bulk of the core and structural ECM being produced by the stroma, while tumour cells express matrix associated components and matrix modulators[62].

One question arising from these findings is whether Endo180-expressing cells in the tumour stroma represent a suitable therapeutic target. An Endo180 targeting antibody drug conjugate (ADC) has been investigated in mice inoculated with human Endo180-positive U937 cells and shown antitumour activity with no discernible toxicity of normal tissue[63], however it is unclear whether this ADC effectively recognises mouse Endo180, hence its utility as stromal targeting agent remains to be established. Consequently, additional reagents, including a fibroblast-specific Endo180 knockout mouse, will be required to fully assess the efficacy of such an approach and potential on-target toxicities in normal tissues. Most importantly, when considering strategies for targeting the tumour stroma, these findings highlight the significance of limiting stromal activation early in the metastatic process.

## Methods

**Reagents and cells**. Antibodies, and the dilutions used, are detailed in Supplementary Table 3. TGFβ1 (R&D Systems). 4T1, 4T07, MRC5, MDA-MB-453, and NIH-3T3 cells were from Isacke laboratory stocks. D2A1 cells were from Ann Chambers laboratory stocks[64]. The generation of the metastatic D2A1-m2 subline has been described previously[37]. The D2A1-m12 subline was generated as described for the D2A1-m2 subline except that the in vivo selection was in BALB/c Endo180$^{-/-}$ mice. In brief, parental D2A1 cells were inoculated into the 4th mammary fat pad of a BALB/c Endo180$^{-/-}$ mouse, the lungs removed at necropsy, dissociated and placed into culture. Tumour cells that grew out were expanded and inoculated into the tail vein of a recipient Endo180$^{-/-}$ mouse and 11–13 days later, lungs were removed at necropsy. In total three rounds of intravenous inoculation were performed. E0771 cells were purchased from CH3 BioSystems. 4T1, D2A1, and E0771 cells were luciferase transduced with lentiviral expression particles containing a firefly luciferase gene and a blasticidin-resistance gene (Amsbio, LVP326). Where indicated, 4T1-Luc cells were transduced with lentiviral particles expressing H2B-mRFP as previously described[48] as well as D2A1, D2A1-m12, and MDA-MB-453 cells with a luciferase2-mCherry vector (Luc2-mCh). mCherry+/RFP+ cells were enriched by fluorescence-activated cell sorting (FACS).

Normal mammary fibroblasts (NMFs) were isolated from the 4th mammary fat pads of naïve 6–8-week old female BALB/c Ub-GFP mouse (BALB/c mice expressing GFP under the human ubiquitin C promoter[65]). Mammary fat pads were cut into small chunks and cultured in DMEM (Dulbecco's modified Eagle's medium) plus 10% FBS (foetal bovine serum), 1% penicillin/streptomycin, and 1% ITS (insulin-transferrin-selenium; Invitrogen). After 7 days, mammary fat pad chunks were discarded, and migratory fibroblasts established in culture. Immortalised NF#1 mouse mammary gland fibroblasts were obtained from Fernando Calvo and have been described previously[39,48]. CAFs were isolated from a 4T1-tumour-bearing BALB/c Ub-GFP mouse. In brief, $1 \times 10^4$ 4T1-Luc-RFP cells were injected into the 4th mammary fat pad of a BALB/c Ub-GFP mouse. After 29 days the primary tumour was resected and homogenised using a McIlwain Tissue Chopper (Campden Instruments) and digested in L-15 medium containing 3 mg mL$^{-1}$ collagenase type I at 37 °C for 1 h, followed by digestion with 0.025 mg mL$^{-1}$ DNase (Sigma) at 37 °C for 5 min. After erythrocyte lysis using Red Blood Cell Lysis Buffer (Sigma), single cells were re-suspended at a density of $1–2 \times 10^7$, and subjected to magnetic sorting to exclude immune cells using rat-anti-mouse CD45 and CD24 antibodies and magnetic sheep-anti-rat Ig Dynabeads (Invitrogen, #11035). FACS for FITC+/RFP−/DAPI− was used to obtain GFP +ve CAFs and

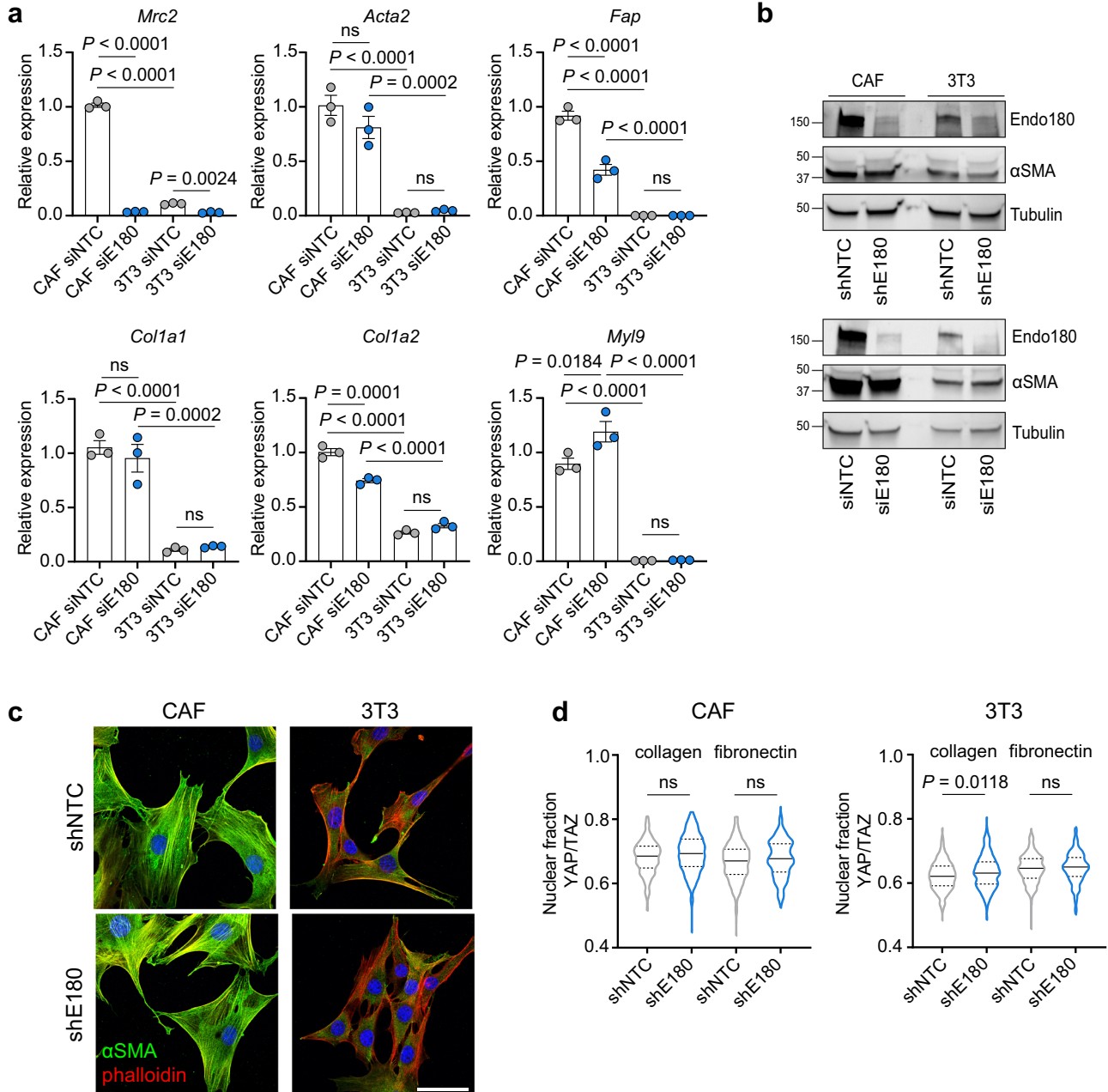

**Fig. 5 Endo180-deficient fibroblasts retain markers of activation. a** RT-qPCR analysis of contractility and activation markers in siNTC and siE180 transfected CAFs and 3T3 cells ($n = 3$, mean values ±SEM, one-way ANOVA, Sidak post-hoc test). **b** Western blot analysis of Endo180 and αSMA in shNTC/shE180 transduced (top panel) and siNTC/siE180 transfected (bottom panel) CAFs and 3T3s. Molecular size markers are in kDa. **c** Confocal images of shNTC and shE180 transduced CAFs (left panel) and 3T3s (right panel) stained for αSMA (green), phalloidin (red) and DAPI (scale bar, 50 μm). **d** siNTC and siE180 transfected CAFs plated on collagen-coated or fibronectin-coated glass 24-well plates for 24 h. YAP/TAZ immunofluorescence intensity was quantified in nuclear and cytoplasmic compartments by high content imaging analysis. Violin plots, solid lines indicate median expression levels, dotted lines indicate upper and lower quartiles ($n = 400$, Kruskal–Wallis test, Dunn's correction). Non-significant (ns) $P$ values > 0.05. Source data are provided as a Source Data file.

to exclude RFP +ve tumour cells and dead cells, respectively. CAFs were maintained in DMEM plus 10% FBS, 1% penicillin/streptomycin and 1% ITS.

shRNA Endo180 knock-down clones from 3T3 and CAFs were generated using lentiviral particles (Sigma, Supplementary Table 4). For siRNA knockdown, two non-targeting controls and two Endo180 targeting oligonucleotides (Supplementary Table 5) were used either separately or as pools in a 1:1 ratio.

All cells were routinely subject to mycoplasma testing.

**In vitro studies**

*Conditioned media (CM).* CM was generated by culturing fibroblasts or tumour cells at a 70–80% confluence. Cells were washed two times and then cultured in serum-free DMEM. After 24 h, CM was collected, centrifuged at 300×*g* and filtered through a 0.2 μm pore filter. Where indicated, CM was supplemented post-collection with FBS for long-term assays.

*Cell proliferation/viability assays.* $1 \times 10^3$ cells/well were seeded into 96-well plates. Cell viability was quantified either by CellTiter-Glo (Promega) at the indicated time points or by time-lapse imaging and quantification of cell confluence using the Live-Cell Analysis System IncuCyte (EssenBioscience).

*Colony formation assay.* Fifty tumour cells were seeded into a 6-well plate and either co-cultured with 5000 fibroblasts, treated with fibroblast conditioned media

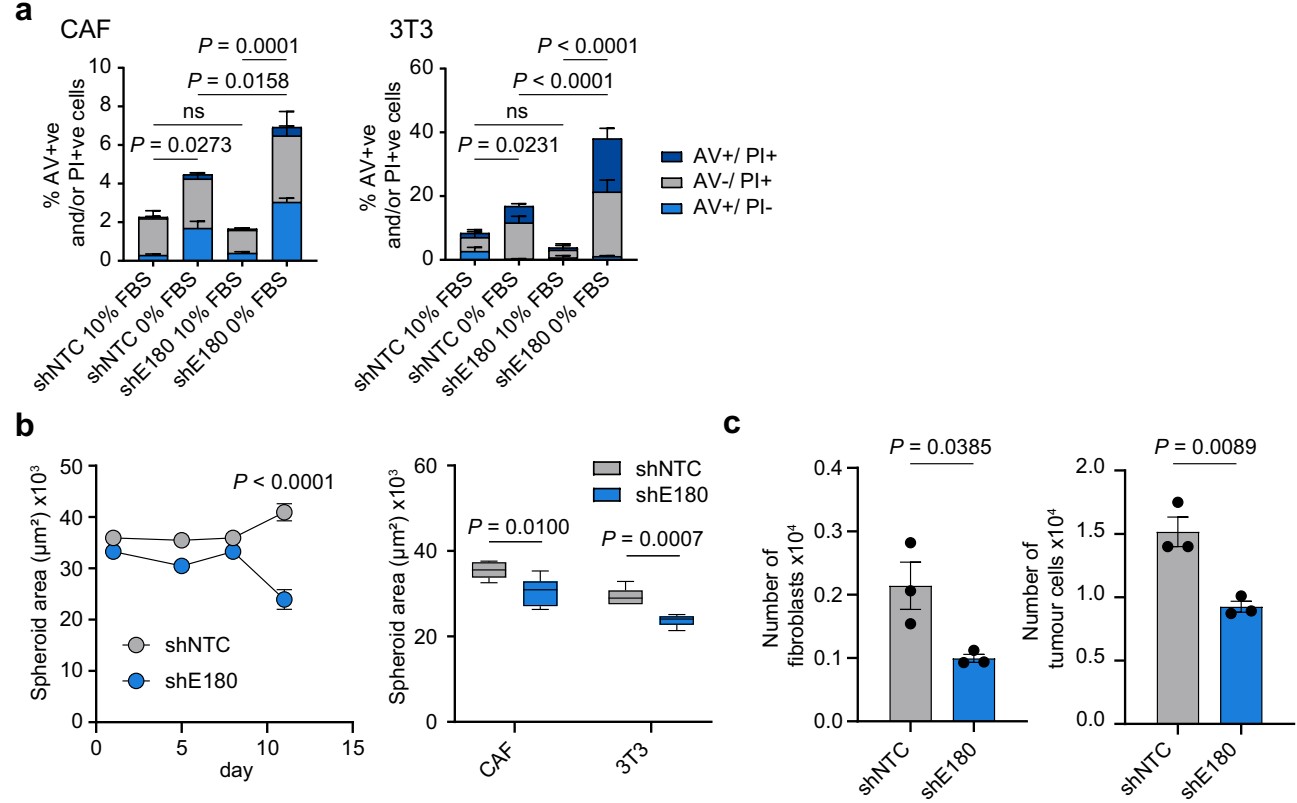

**Fig. 6 Reduced viability of Endo180-deficient CAFs. a** shNTC and shE180 CAF and 3T3 cells were plated in anoikis conditions (flat bottom low adherence 6-well plates) in DMEM plus 0 or 10% FBS. After 24 h level of cell death was measured by Annexin V (AV)/ PI staining (*n* = 3 per condition; mean values ± SEM, one-way ANOVA of total % AV +ve and/or PI +ve cells). **b** Left panel, 900 shNTC or shEndo180 CAFs were plated into U-bottom wells and growth of suspension aggregates size monitored over time (*n* = 6; mean values ± SEM, two-way ANOVA). Right panel, 900 CAFs or 3T3 fibroblasts were plated into U-bottom wells and suspension aggregate size measured on day 10. Box plots shows median and 25th–75th quartiles, whiskers show minimum and maximum. (*n* = 6; mean values ± SEM, two-sided *t*-test). **c** 300 D2A1 and 600 GFP-CAFs cells were co-cultured in U-bottom plates. After 8 days, spheroids were dissociated and the number of GFP-positive CAFs and GFP-negative tumour cells counted (*n* = 3 per condition; mean values ± SEM, two-sided *t*-test). Non-significant (ns) *P* values > 0.05. Source data are provided as a Source Data file.

or plated onto fibroblast-derived ECM. Tumour colonies were stained 7 days later with crystal violet (Sigma). Plates were scanned using the GelCount (Oxford Optronix) and image analysis performed using GelCount software and FIJI.

*Spheroid assays.* Tumour cells and fibroblasts were mixed in a 1:9, 1:4, or 1:2 ratio and seeded into ultralow adherence 96-well U-bottom plates (Corning). Spheroid growth was monitored using the Celigo Image Cytometer (Nexcelom Bioscience), and cell viability was monitored with CellTiter-Glo at the indicated time points. To ensure proper lysis of the spheroids, the incubation time with the CellTiter-Glo reagent was extended from 10 to 30 min before recording luminescence. As indicated cases, spheroids containing D2A1 cells and GFP-fibroblasts were dissociated using Accumax solution (Sigma) and cell numbers were quantified using the Celigo Image Cytometer (Nexcelom Bioscience) and a fluorescent Countess II FL Automated Cell Counter (Invitrogen). For CM-spheroid assays 200 D2A1 cells were seeded in DMEM supplemented with 10% FBS. After 2 days spheres were collected, washed with serum free media and replated in CM supplemented with 2% FBS. Spheroid area was quantified after 8 days culture in CM.

*Generation of extracellular matrices (ECM).* Twenty-four-well plates were coated with 0.2% gelatin (Sigma, G1393) and cross-linked with 1% glutaraldehyde/PBS. Cross-linking was quenched with 1 M glycine/PBS. Fibroblasts were cultured in DMEM or tumour cell conditioned media supplemented with 2% FBS and 50 µg mL$^{-1}$ ascorbic acid ascorbic acid (Sigma, A4403) on gelatin-coated plates. The medium was changed every 2 days for 7 days and fibroblasts were lysed using pre-warmed extraction buffer per well (20 mM NH$_4$OH, 0.5% Triton X-100 in PBS). Residual DNA was digested with 10 µg mL$^{-1}$ DNase I (Sigma). Matrices were either fixed with 4% paraformaldehyde and stained for fibronectin or used in growth assays. Confocal z-stacks were acquired using the ImageXpress Micro Confocal High-Content Analysis System (Molecular Devices) equipped with a 60 µm pinhole spinning-disk, and a plan apo λ 10×/0.45 NA objective (Nikon). Z-stacks were maximally projected and analysed with ImageJ1.52p and the

"TWOMBLI" plugin[38], to quantify matrix density and (an-)isotropy of fibres. Data shown are matrix density, calculated as (100 − HDMvalue × 100) to convert from high density matrix background to % fibre density. End and branching points were normalised by dividing the raw value by the total length of the fibres. Alternatively, matrices in 24-well plates were extensively washed with PBS before tumour cells were seeded for a colony formation assay (50 cells per well) or IncuCyte imaging as described above (4 × 10$^3$ cells per well).

*Immunofluorescence staining.* Fibroblasts were cultured overnight on coverslips, starved overnight in serum-free DMEM and then treated with or without TGFβ1 (5 ng mL$^{-1}$, R&D Systems) for 1 h before fixation with 4% paraformaldehyde, permeabilisation with 0.5% Triton X-100 and staining with DAPI, Alexa dye-labelled phalloidin, and anti-p(Ser19)-MLC or anti-αSMA antibodies. Cells were imaged on a Leica TCS SP8 confocal microscope.

*Cell shape and YAP/TAZ distribution analysis.* 1 × 10$^3$ cells/well were seeded into a ViewPlate-96 Black with optically clear bottom (PerkinElmer) or Softwell 24, Easy Coat plates (Cell guidance system) with the indicated stiffness. Plates or glass coverslips were coated with 5 g cm$^{-2}$ rat-tail collagen I (Corning, #354236) or fibronectin (R&D, 1918-FN-02M). After 24 h, cells were fixed in 4% paraformaldehyde, permeabilised with 0.5% Triton X-100 prior to staining with DAPI, Alexa dye-labelled phalloidin and, where indicated, anti-YAP antibody. Automated image acquisition was performed on an Operetta high content imaging system (Perkin Elmer). Cell shapes and cytoplasmic and nuclear YAP/TAZ distribution were analysed using basic algorithms in the Harmony high content analysis software package (Perkin-Elmer). Cells were initially defined using the DAPI channel to identify the nucleus and the cytoplasm was segmented using the Alexa488/Alexa555 channel. After this, the Harmony software allows the extraction of cell shape parameters, including "cell roundness".

*Fibroblast contraction assays.* 7 × 10$^4$ to 1 × 10$^5$ fibroblasts embedded in rat tail collagen I (Corning; final concentration 2 mg mL$^{-1}$) were added to 24-well plates

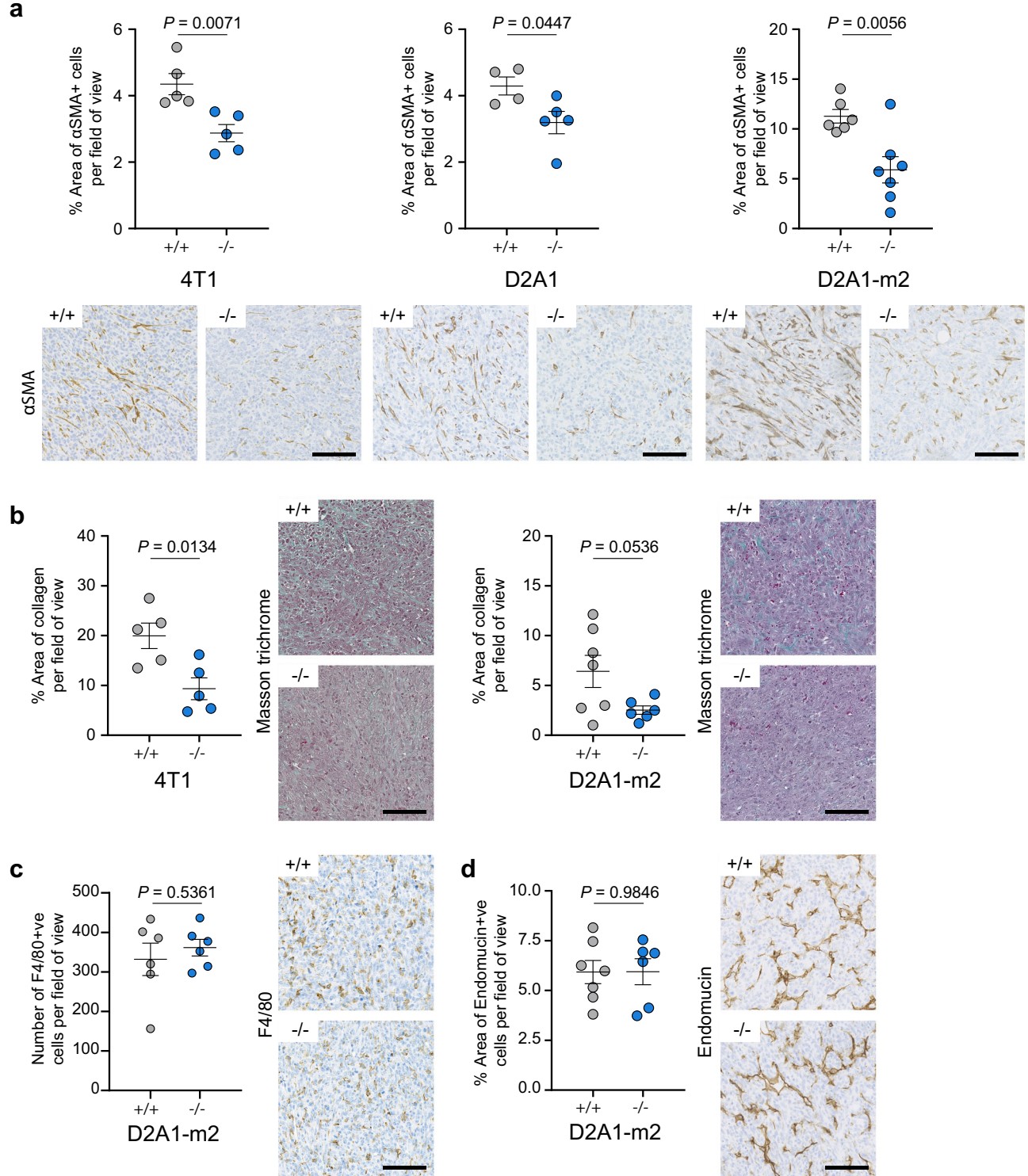

**Fig. 7 Reduced fibroblast numbers and collagen deposition. a** Analysis of intra-tumoural αSMA positive staining in 4T1, D2A1 and D2A1-m2 primary tumours grown in Endo180+/+ and Endo180−/− mice. Data represents mean values per tumour ± SEM (4T1, $n = 5$ per group; D2A1, Endo180+/+ $n = 4$, Endo180−/− $n = 5$; D2A1-m2, Endo180+/+ $n = 6$, Endo180−/− $n = 7$; two-sided $t$-test) Representative images (scale bar, 100 μm). **b** Analysis of intra-tumoural Masson's trichrome staining of 4T1 (left panel) and D2A1-m2 (right panel) primary tumours in Endo180+/+ and Endo180−/− mice. Data represent mean values per tumour ±SEM (4T1, $n = 5$ per group; D2A1-m2, Endo180+/+ $n = 7$, Endo180−/− $n = 6$; two-sided $t$-test). Representative images (scale bar, 100 μm). **c** Analysis of intra-tumoural F4/80 staining of D2A1-m2 primary tumours in Endo180+/+ and Endo180−/− mice. Data represent mean values per tumour ±SEM ($n = 6$ per group; two-sided $t$-test). Representative images (scale bar, 100 μm). **d** Analysis of intra-tumoural endomucin staining of D2A1-m2 primary tumours. Data represent mean values per tumour ±SEM (Endo180+/+ $n = 7$, Endo180−/− $n = 6$; two-sided $t$-test). Representative images (scale bar, 100 μm). Non-significant (ns) $P$ values > 0.05. Source data are provided as a Source Data file.

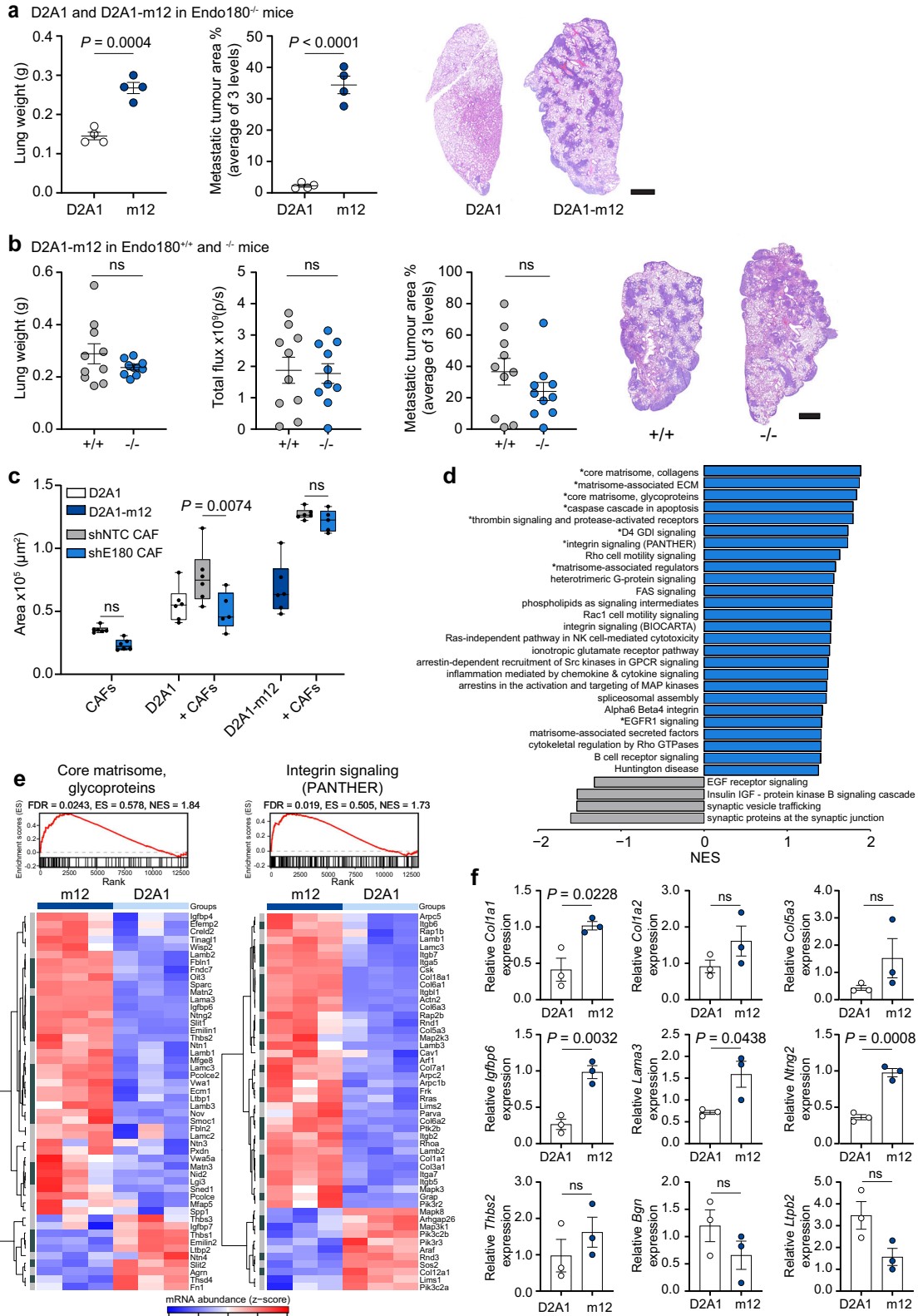

and incubated at 37 °C. After the gels were set, media or conditioned media was added, as indicated. 24–72 h later, plates were scanned and the contracted gel area was measured using FIJI.

*Invasion assays.* $1 \times 10^3$ fibroblasts were seeded into ultra-low adherence 96-well U-bottom plates. 24 h later, fibroblast aggregates were embedded into 100 μL 2 mg mL$^{-1}$ rat tail collagen I. Invasion into collagen was imaged at the indicated time points and analysed using FIJI.

*Apoptosis assay.* $5 \times 10^4$ cells/well were plated into either a tissue culture or low adherence 6-well plates. Twenty-fours hour after seeding, cells were stained with the Annexin V-APC/ PI (propidium iodide) Apoptosis Detection Kit (eBioscience) and analysed using a BD Biosciences LSRII flow cytometer with FACSDIVA and FlowJo software.

*Western blotting.* Cells were lysed in 4× Laemmli sample buffer (Bio-Rad) and sonicated. Cell lysates were subject to immunoblotting with the Bio-Rad Western

**Fig. 8 Selected tumour sublines can overcome a defect in stromal Endo180. a** $4 \times 10^5$ D2A1-Luc2-mCh or D2A1-m12-Luc2-mCh cells were injected intravenously into BALB/c Endo180$^{-/-}$ mice ($n = 4$ mice per group) and the experiment was terminated on day 12. Data represent mean values ± SEM. Shown are ex vivo lung weights ($t$-test), quantification of metastatic tumour area in the lungs (two-sided $t$-test) and representative H&E stained lung sections (scale bar, 1 mm). **b** $4 \times 10^5$ D2A1-m12-Luc cells were injected intravenously into BALB/c Endo180$^{+/+}$ or Endo180$^{-/-}$ mice ($n = 10$ per group) and the experiment terminated on day 12. Data represent mean values ± SEM. Shown are ex vivo lung weights (two-sided $t$-test), ex vivo IVIS imaging of the lungs (two-sided $t$-test), quantification of % metastatic tumour area in the lungs (two-sided $t$-test) and representative H&E stained lung sections (scale bar, 1 mm). **c** Tumour cells (D2A1 or D2A1-m12) and/or shNTC/shEndo180 CAFs were cultured alone or co-cultured in U-bottom plates and spheroid size measured at day 8 ($n = 6$ spheroids per group, 2 way-ANOVA, Sidak post-hoc test). **d** fGSEA, upregulated (blue) and downregulated (grey) in D2A1-m12 compared with D2A1 cells. All pathways displayed had $P$ values < 0.05 and those labelled with a * had FDR adjusted $P$ value < 0.1. NES normalised enrichment score. **e** fGSEA of the pathways: core matrisome, glycoproteins, and integrin signalling pathway (PANTHER); and associated heatmap showing the genes of the pathway ($n = 3$; heatmap scale is a $z$-score). Dark grey, significant DEGs with |log$_2$FC|> 1 and adjusted $P$ value ≤ 0.05. Light grey, non-significant DEGs. **f** RT-qPCR analysis of matrisome genes in D2A1 and D2A1-m12 cells isolated from in vivo primary tumours ($n = 3$ tumours per group, mean values ± SEM, two-sided $t$-test). Non-significant (ns) $P$ values > 0.05. Source data are provided as a Source Data file.

blot system according to manufacturer's protocol. Blots were blocked with 5% BSA before incubation with primary and secondary antibodies.

**In vivo procedures**. All animal work was carried out under UK Home Office Project Licences 70/7413 and P6AB1448A granted under the Animals (Scientific Procedures) Act 1986 (Establishment Licence, X702B0E74 70/2902), and was approved by the "Animal Welfare and Ethical Review Body" at The Institute of Cancer Research (ICR). Mice with a genetic deletion in Endo180 (*Mrc2*)[15] were backcrossed for at least six generations with either BALB/c or C57BL/6 (Charles River) mice. Genotypes were confirmed by PCR. Endo180$^{-/-}$ colonies were maintained at the ICR. Aged-matched 6–8 week-old female BALB/c or C57BL/6 mice were purchased from Charles River. Animals were housed in IVC type cages which are run under negative airflow. Mice were companion held, had food and water ad libitum and monitored daily by ICR Biological Services Unit staff. Animal holding rooms were maintained within the parameters recommended in the Home Office Code of Practice with temperatures being 21 °C +/− 2 °C, humidity 55% +/− 10% and a light cycle of 12 h dark/light. In all cases, experiments were terminated if the primary tumour reached a maximum allowable diameter of 17 mm or if a mouse showed signs of ill health.

*Orthotopic inoculation.* 1–5 × 10$^4$ 4T1, D2A1, D2A1-m2, or D2A1-m2 cells were injected into the 4th mammary fat pad under general anaesthesia. Tumour growth was measured twice a week using callipers up to a maximum diameter of 17 mm. Tumour volume was calculated as $0.5236 \times [(\text{width} + \text{length})/2]^3$.

*Intravenous inoculation.* 1 × 10$^5$ 4T1-Luc, 2.5 × 10$^5$ 4T07, 4 × 10$^5$ D2A1, D2A1-m2-Luc, or 4 × 10$^5$ E0771-Luc cells were injected into the lateral tail vein of mice. Mice were sacrificed when the first animal showed signs of ill health.

*Intrasplenic inoculation.* 1 × 10$^5$ 4T1-Luc cells were inoculated into the spleen parenchyma of BALB/c mice under general anaesthesia.

Unless otherwise stated, primary tumours and lungs were weighed at necropsy. For IVIS imaging, mice were injected intraperitoneally with 150 mg kg$^{-1}$ D-luciferin (Caliper Life Sciences) in 100 µL and mice imaged in vivo using an IVIS imaging chamber (IVIS Illumina II). Luminescence measurements (photons/second/cm$^2$) were acquired over 1 min and analysed using the Living Image software (PerkinElmer) using a constant size region of interest over the tissues. Alternatively, 5 min after D-luciferin injection, dissected lungs, or livers were imaged ex vivo and quantified as total flux values from organs.

Where indicated, primary tumours were dissociated using a mouse tumour dissociation kit (130-096-730, Miltenyi Biotec) with the 37C_m_TDK_2 programme on the gentleMACS Octo Dissociator (Miltenyi Biotec). The resulting cell pellet was resuspended in Red Blood Cell Lysis buffer (Sigma) for 5 min, spun and resuspended in FACS buffer (PBS with 5% FBS) containing a 1:100 dilution of anti-mouse CD16/CD32 block for 10 min at room temperature. Cells were stained with directly conjugated CD45, CD31, F4/80 and PDGFRα antibodies for 30 min at 4 °C. Cells were co-stained with DAPI (1:10,000) to exclude dead cells, washed with PBS and sorted on a FACS Aria III. Sorted populations were defined as tumour cells (RFP+), fibroblasts (PDGFRα+), endothelial cells (PDGFRα−), macrophages (CD45+/F4/80+), other immune cells (CD45+/F4/80−) and negative for all other markers.

*Quantification of metastatic burden and immunohistochemistry.* Three to four micrometer lung and/or liver FFPE sections, approximately 150 µm apart, were cut and stained with haematoxylin and eosin (H&E). Sections were scanned using the NanoZoomer Digital Pathology and file names blinded. Total number of individual nodules was counted manually in three sections, per animal. Lung metastatic area was quantified as the mean % tumour area per lung section or by counting macroscopic tumour nodules counted manually. FFPE sections of primary mouse tumours were stained with αSMA, endomucin, F4/80 or Masson's trichrome and

detection was achieved with the VectaStain ABS system. Stained sections were scanned on the NanoZoomer Digital Pathology (Hamamatsu). Images were exported and analysed with ImageJ and NDPITools ImageJ plugin[66]. HRP staining was analysed in ImageJ from ≥6 random intra-tumoural fields of view per tumour section, avoiding areas of necrosis. HRP images were colour deconvoluted using the ImageJ H DAB vector and converted into 8-bit images and the % area or number of cells quantified (threshold, 0–130). Intra-tumoural collagen content in Masson's trichrome stained sections was analysed similar to HRP staining, using the ImageJ Masson's trichrome Vector.

**RT-qPCR**. RNA was isolated using Qiagen RNeasy kit and cDNA was generated using the QuantiTect reverse transcription kit (Qiagen) according to the manufacturer's instruction. RT-qPCR was performed with Taqman Gene Expression Assay probes (Supplementary Table 6) on an ABI Prism 7900HT, and relative quantification was performed using QuantStudio Real-time PCR software. Each reaction was performed in triplicate. Relative expression levels were normalised to *B2m/B2M* or *Ubc/UBC* endogenous control, with a confidence interval of 95% for all assays. For co-culture Endo180 (*MRC2*) expression analysis, MRC5 lung fibroblasts were directly co-cultured with either D2A1-RFP or MDA-MB-453-RFP cells for 48 h before FACS isolation of RFP-negative fibroblasts

**RNA-Seq analysis**. RNA from D2A1 and D2A1-m12 cells ($n = 3$ per line) was extracted using the RNeasy kit according to the manufacturer's instructions. Quality and quantity of RNA were assessed using a Qubit and Bioanalyzer. NEBNEXT Ultra II Directional RNA and polyA RNA selection kits were used to generate libraries, which were combined and sequenced at PR 100 cycles on a NovaSeq flowcell. All samples were run to achieve ~50 million clusters. RNA-Seq generated 11.4–53.1 million reads per sample. FastQC (v0.11.4) was used to evaluate the library quality. Paired-end reads (100 base pair long) were aligned to the mouse reference genome GRCm38, using STAR v2.5.1b[67] with—quantMode GeneCounts and—twopassMode Basic alignment settings. Annotation file used for feature quantifications was downloaded from GENCODE (v17) in GTF file format. Post alignment quality control was performed using RseQC (v2.6.3)[68].

Differential mRNA abundance analysis was performed using edgeR package (v3.22.5)[69] in R (v3.5.0) using the model ~0 + group. Genes with low expression were filtered out by retaining genes with count-per-million (CPM) counts >1 in at least two samples. Samples were normalised using edgeR's TMM (trimmed mean of *M*-values) method and differential expression was determined using quasi-likelihood (QL) *F*-test. Results were further annotated using ENSEMBL gene annotations from R package org.Mm.eg.db (v3.6.0). Genes with |log$_2$FC|>1 and adjusted $P$ value ≤ 0.05 were considered statistically significant.

Fast Gene Set Enrichment Analysis (fGSEA) was performed with R package fgsea (v1.8.0) using mouse gene sets (Biocarta, NetPath, Panther) downloaded from GSKB (Gene Set Knowledgebase in Mouse) database, and the mouse gene sets obtained from the Matrisome Project (matrisomeproject.mit.edu). Genes were ranked as: − log$_{10}$(P) × sgn(log$_2$FC). We applied a minimum gene set size of ten genes and performed the analysis using 10,000 permutations. We considered pathways with normalised enrichment score (NES) > 0 and $P$ value < 0.05 to be upregulated and pathways with NES < 0 and $P$ value < 0.05 to be downregulated.

**Bioinformatics analysis**. For the publicly available datasets used, and their access, see "Data availability" section.

For association of Endo180 (*MRC2*) gene expression in TCGA and NKI dataset with fibroblast TGFβ response signature (F-TBRS), the gene set was retrieved from https://doi.org/10.1016/j.ccr.2012.08.013[32]. There were 286 probes representing 171 unique annotated genes. In the TCGA dataset 151 out of the 171 F-TBRS genes were matched. Average gene expressions of the 151 genes from TCGA dataset was used as signature score for each tumour.

Single cell RNA-Seq data of CAFs isolated from MMTV-PymT tumours was downloaded from the published study[7]. Briefly, single end sequencing reads data

was aligned to the mouse genome (assembly: mm10) using STAR aligner (v2.3.0). Aligned reads were quantified and normalised to genes using the Reads Per Kilobase of transcript, per Million mapped reads (RPKM) measurement. Endo180 (*Mrc2*) gene expression levels $\log_2$(RPKM + 1 are clustered according to identified populations in the original publication (vascular CAFs, matrix CAFs, developmental CAFs, and actively cycling CAFs).

**Statistics and reproducibility**. Statistics were performed using GraphPad Prism 8. Unless otherwise indicated, all comparisons between two groups were made using two-tailed, unpaired Student's *t*-test. If the analysis did not pass normality test (Shapiro–Wilk test) groups were analysed by Mann–Whitney *U*-test. If more than two groups were compared One-way ANOVA analysis was performed with Tukey test for multiple comparison. Where multiple groups with a second variable, e.g., over time, were compared, a two-way ANOVA followed by Sidak post-hoc testing was performed. $*P < 0.05$; $**P < 0.01$; $***P < 0.001$. Box plots show median, 25th–75th quartiles, whiskers show minimum and maximum.

Animal experiments were repeated on at least one additional occasion with similar results except experiments shown in Fig. 1f, i, and j which were conducted once. All in vitro experiments were repeated on at least one additional occasion with similar results or validated with different cell lines. Representative data from micrographs are as follows: Fig. 1a, illustrative examples of 2 cores from a 254 core tissue microarray; Fig. 2b, repeated in two independent mouse experiments, each with $n = 5$ mice per group; Fig. 3b, repeated in >4 independent experiments with each having at least four replicas using siRNAs transfected and/or shRNA transduced cells; Fig. 4a, b cell shape analyses was repeated twice using fluorescent labelling, and also observed in brightfield images; Figs. 4f, g and 5b, c Western blot experiments were repeated on two additional occasions with similar results; Fig. 7a–d are representative images out of at least six fields of view and at least one additional occasions with similar result.

**Reporting summary**. Further information on research design is available in the Nature Research Reporting Summary linked to this article.

## Data availability

The RNA-Seq data for this study (Fig. 8) have been deposited in the European Nucleotide Archive (ENA) at EMBL-EBI under accession number PRJEB36901. The whole exome sequencing data (Supplementary Fig. 8c–e) is available under the accession number PRJEB43908. Publicly available data can be accessed as follows: Series matrix files for TCGA 522 primary breast cancer samples (Fig. 1g) were downloaded from [https://tcga-data.nci.nih.gov/docs/publications/brca_2012/]. NKI data (Fig. 1g) were retrieved from breastCancerNKI R package (http://bioconductor.org/packages/release/data/experiment/html/breastCancerNKI.html)[70]. The following datasets were retrieved from the Gene Expression Omnibus (GEO) site: GSE12622[34] (Fig. 1c): GSE41678[71] Fig. 1e); GSE37614[71] (Fig. 1f); GSE111229[7] (Fig. 1h) GSE39397 (GPL13158)[32] (Supplementary Fig. 1c). scRNASeq melanoma data[33] (Supplementary Fig. 1d, e) (https://www.ebi.ac.uk/gxa/sc/experiments/E-EHCA-2/downloads) visualised with https://melanoma.cellgeni.sanger.ac.uk/ and lung cancer data (Supplementary Fig. 1f) PMID29988129[36] (https://gbiomed.kuleuven.be/scRNAseq-NSCLC) visualised with SCope (https://scope.aertslab.org/). The remaining data are available within the article, Supplementary Information, Source Data file or available from the authors upon request. Source data are provided with this paper.

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

## Acknowledgements

This work was funded as part of Programme Funding to the Breast Cancer Now Toby Robins Research Centre (C.M.I.), Worldwide Cancer Research (15-0355, C.M.I. and U.J.) and a Schrödinger fellowship of the Austrian Science Fund (FWF) (J3434-B13, U.J.). We acknowledge NHS funding to the NIHR Biomedical Research Centre at the Royal Marsden and the ICR. We would like to thank Adam Mills for the whole exome sequencing analysis, Kristian Pietras and Jonas Sjölund for the data analysis and images in Fig. 1h, Mirjana Efremova help in accessing the data shown in Supplementary Fig. 1e, Miriam Melake for help with the matrix imaging, Gernot Walko for his imaging expertise and scientific discussions, Naomi Guppy and her team in the Breast Cancer Now Toby Robins Research Centre Nina Barough Pathology Core Facility for histopathology support, Fredrik Wallberg for help with the FACSorting and imaging, and the following facilities at the ICR; Tumour Profiling Unit, Biological Services Unit, and the FACS and Light Microscopy Facility. The results published here are in whole or part based upon data generated by the TCGA Research Network (http://cancergenome.nih.gov/).

## Author contributions

Conception and design of the work (U.J. and C.M.I.). Acquisition, analysis, interpretation of data (U.J., Av.W., R.E., L.J., D.V., J.A., Q.G., S.H., M.I., and C.M.I.). Preparation of the manuscript (U.J., S.H., M.I., and C.M.I. with input from all other authors)

## Competing interests

The authors declare no competing interests.
