## [Peer Review File · Nature Communications]

REVIEWER COMMENTS

Reviewer #1 (Expertise: CAFs, breast cancer metastasis, Remarks to the Author):

In this study by Jungwirth et al. the authors investigated the role of CAF-derived Endo180 in facilitating growth and metastasis of breast cancer. By analyzing several available datasets of CAF sc-RNAseq expression data, they show that Endo180 is predominantly expressed by a population of matrix-remodeling CAFs. They show in vivo that host-derived Endo180 is functionally important for tumor growth and metastasis, by using multiple cell lines of breast cancer and several routes of inoculation in Endo180^{-/-} mice. In vitro, they used fibroblasts in which the expression of Endo180 was knocked down and showed that loss of Endo180 leads to a fibroblast impairment in contractility and reduced CAF viability, suggesting that this is the mechanism by which it promotes tumor growth. Finally, in an interesting and elegant experiment the authors selected for a tumor cell line that overcame these stromal defects by in vivo passages in Endo180^{-/-} mice and found that these cells upregulated matrix-related gene expression.

Overall, the study is solid and interesting. While it does not provide any novel concepts or paradigms regarding the role of CAFs in cancer, the study elucidates a mechanism by which a protein expressed preferentially by matrix-remodeling CAFs facilitates tumor growth.

The data is mostly convincing and of high quality. A main weakness of the paper is that the in vivo experiments do not actually provide information about the specific role of CAFs, but rather of the entire microenvironment as they were performed in mice in which Endo180 was KO in all tissues.

Specific Comments

1. Figure 1C: The TCGA dataset includes expression from total tumor tissue, so it does not really allow any conclusions regarding expression in CAFs. It would be better to perform this analysis in datasets that separate tumor from stromal expression (e.g. Finak G et al. Nature Med. 2008).

2. Figure 2,3: The authors tested the functional importance of Endo180 for tumor growth and metastasis, using multiple cell lines and genetic backgrounds, as well as different routes of inoculation. However, the targeted deletion in Endo180 is not cell type specific, and thus all the host cells in these mice will have the deletion, including immune cells, endothelial cells etc. While there is currently no good specific way to genetically target gene expression in fibroblasts in vivo, the authors should at least acknowledge and discuss this limitation. There is also a way to deal with this experimentally, at least in the primary tumor assays. The authors should substantiate their claim

that CAF-derived Endo180 is functionally important by co-injecting orthotopically tumor cells mixed with Endo180^{-/-} fibroblasts (from knockout mice), compared with WT fibroblasts. Currently, the authors' conclusion from these experiments that "Together these data provide strong evidence that upregulated expression of Endo180 in matrix CAFs is required for efficient tumour progression in vivo" is very overstated, considering that the deletion is general, and not specific to matrix CAFs.

3. The title "In vitro assays recapitulate in vivo phenotypes" of the section regarding Figure 4 (p. 7) is not very informative. The in vivo data in figures 2,3 implied a functional role for host-derived Endo180 in facilitating tumor progression. The in vitro data in figure 4 only recapitulates an observation on the effect of fibroblast-expressed Endo180 on tumor cell proliferation in 3D. The other in vitro experiments show that effect is not mediated by secreted factors or by the matrix.

4. Minor comment: there are some typos along the manuscripts that should be fixed.

Reviewer #2 (Expertise: scRNAseq, Remarks to the Author):

In this manuscript, Jungwirth and colleagues investigate the impact of Endo180-expressing fibroblasts on tumor growth and metastasis. Single-cell and bulk expression profiling of cancer-associated fibroblasts (CAFs) from different tumor types indicate preferential expression of MRC2 (Endo180) on CAFs. As detailed below, there are several major concerns related to the scRNA-seq aspects of this study:

Major comments:

1) The in vivo conclusions in this manuscript rely on the assumption that MRC2 (Endo180) expression is specific to fibroblasts. While the immunostaining and publicly available expression data in this manuscript indicate a strong enrichment of MRC2 on fibroblasts, these data, especially the scRNA-seq data, are highly selective and insufficient to make the authors' case. In perusing MRC2 expression across multiple scRNA-seq tumor atlases (<http://tisch.comp-genomics.org/search-gene/?genesearch=MRC2>), MRC2 appears to be variably expressed on malignant cells and myeloid subsets (e.g., macrophages), among other cell types, in various cancers. Whether this expression is translated into productive Endo180 protein products cannot be determined from scRNA-seq data, but at a minimum, these data warrant tempered claims about the specificity of MRC2 for CAFs, particularly in the absence of additional experiments. This is especially critical since the in vivo knockdown experiments in this work presume that the phenotypes observed can be specifically

traced to CAFs, but M2-polarized macrophages are also implicated in tumor growth and metastasis in various solid tumors.

2) Related to the above, it would be ideal if the authors would sort and profile MRC2 (Endo180) expression on major tumor-associated lineages, including macrophages, in WT and Endo180 KO mice to demonstrate specificity of MRC2 for CAFs within their *in vivo* system.

3) The authors provide no methodological details about the scRNA-seq data processing and visualization steps used in Fig. 1h and Supplementary Fig. 1c, making it difficult to properly assess these data.

4) The statistical significance of the result in Fig. 1h (right) is missing.

Minor comments:

1) RPKM is not commonly used to represent scRNA-seq expression data. TPM (transcripts per million) or CPM (counts per million) is preferred.

2) The authors jump from one cancer type (breast, melanoma, colorectal) and species (human/mouse) to another in the bioinformatics analyses on pages 4 and 5 without a clear/compelling rationale or narrative.

Reviewer #3 (Expertise: Matrix remodeling, cancer metastasis, Remarks to the Author):

The manuscript by Jungwirth et al. uses both *in vitro* and *in vivo* approaches to dissect the role of stromal cell (fibroblast) Endo180 in the growth of breast cancer cell lines at both primary and secondary metastatic sites. The manuscript is well written, and conclusions are generally supported by the data provided. This work would certainly be of interest to the field and based on the authors addressing my comments below I would recommend publication.

The structure of the manuscript appears to jump back and forth a little from *in vivo* to *in vitro* and back again. For example, staining for Endo180/ α SMA / Collagen may be better situated with the data on tumor growth in Figure 2 rather than in figure 6, as this would be the logical point to ask the question about what was happening to CAF activation and ECM remodeling in the slower growing tumors.

Some major questions that remain to be addressed are:

Does Endo180 KD alter CAF activation?

Where α SMA is decreased in the tumor histology (F6), is this simply a reflection of less stromal cells overall?

Do the D2A1m12 cells have elevated expression of Endo180? It has been previously shown that tumor cells might acquire increased Endo180 expression that plays a role in invasive tumor growth.

It would greatly strengthen the manuscript to see overexpression and/or recovery experiments in the low / KD Endo180 cells to support the authors conclusion that this is an Endo180 causal event.

Monoclonal antibodies to Endo180 have been trialed in a number of settings. Do the authors have any evidence that such an approach (or similar Endo180 targeting approaches) would be relevant to this setting, given the title of their manuscript?

The findings that there is less collagen accumulation in Endo180^{-/-} tumors contradicts earlier work where Endo180 knockout led to less tumor burden and increased collagen accumulation in the tumors as a result of defective collagen clearance by tumor-associated stromal cells. Perhaps the authors can comment on this to put their findings into this context.

In F6 the authors show decreased collagen deposition in tumors. Does the KD of Endo180 in CAFs reduce Col1a1/Col1a2 expression in these cells? This would help with my previous comment.

Other than an underpowered (n=2) bulk contraction assay and a cell spreading assay, the manuscript does not present compelling evidence that the differential activation of contractile machinery is dependent on Endo180 expression, and not as a result of differential CAF activation.

In F5, n=2 plugs is insufficiently powered. Additional repeats should be carried out.

The different stiffnesses of the matrix (F5 and SF6) are known to alter adhesion, fibroblast activation and subsequent contractility. The fact that Endo180 deficient cells remain more spread on a soft matrix does not necessarily confirm they have a contractile dysfunction, if the authors have not

ruled out differential activation or adhesion. The authors should consider staining for markers of activation (αSMA) contractility (pMLC, pMYPT) and stress fibers to support their data.

Given the RNAseq data in F7, the authors should show elevated αSMA in D2A1-m12 cancer cells in vitro in tumor/metastases to support their conclusion. Similarly, some validation of the 9 most significantly enriched collagens / glycoproteins should be undertaken in the in vivo tumors.

As a supplemental table, it could be helpful to include the table of ECM and ECM-related genes and their changes that the authors say Endo180 is one of. This would allow other researchers to see what may be changing alongside Endo180.

On this, how do those genes change in the 'omics analysis the authors present in Figure 7. Is the loss of Endo180 leading to a loss of expression of other "matrix CAF" genes that may also explain the phenotypic effects?

It is not clear why the 3T3 normal fibroblasts were used as the authors state than Endo180 is expressed at a low level in normal stromal fibroblasts.

Some of the discussion, such as the endosialin KO and MT1-MMP knockout does not seem relevant, unless the authors are attempting to incorporate previous work showing that partial cleavage of collagens by MT-MMP1 leads to increased Endo180 binding and internalization of collagens? However, given the authors case for a non-internalization theory, the focus of the discussion could be tightened. Perhaps instead the authors could discuss their findings in the context of the domain structure of Endo180 and give some insight as to which of the dozen or so domains is likely important in the context of this manuscript especially given that the CTLD domains have been shown to be important in binding glycosylated collagens such as those found in the basement membrane which play an important role in metastasis at both primary and secondary sites.

Impairment of a distinct cancer-associated fibroblast population limits tumour growth and metastasis

Jungwirth et al. NCOMMS-20-07141A-Z

Response to reviewer comments

We thank the reviewers for their constructive feedback. Please note, due to the insertion of new data, many of the Figure numbers have changed in this revised manuscript. In our response to reviewer comments, we referred to the new Figure numbers.

Reviewer #1 (Expertise: CAFs, breast cancer metastasis, Remarks to the Author):

In this study by Jungwirth et al. the authors investigated the role of CAF-derived Endo180 in facilitating growth and metastasis of breast cancer. By analyzing several available datasets of CAF sc-RNAseq expression data, they show that Endo180 is predominantly expressed by a population of matrix-remodeling CAFs. They show in vivo that host-derived Endo180 is functionally important for tumor growth and metastasis, by using multiple cell lines of breast cancer and several routes of inoculation in Endo180^{-/-} mice. In vitro, they used fibroblasts in which the expression of Endo180 was knocked down and showed that loss of Endo180 leads to a fibroblast impairment in contractility and reduced CAF viability, suggesting that this is the mechanism by which it promotes tumor growth. Finally, in an interesting and elegant experiment the authors selected for a tumor cell line that overcame these stromal defects by in vivo passages in Endo180^{-/-} mice and found that these cells upregulated matrix-related gene expression.

Overall, the study is solid and interesting. While it does not provide any novel concepts or paradigms regarding the role of CAFs in cancer, the study elucidates a mechanism by which a protein expressed preferentially by matrix-remodeling CAFs facilitates tumor growth.

The data is mostly convincing and of high quality. A main weakness of the paper is that the in vivo experiments do not actually provide information about the specific role of CAFs, but rather of the entire microenvironment as they were performed in mice in which Endo180 was KO in all tissues.

Specific Comments

1.1. Figure 1C: The TCGA dataset includes expression from total tumor tissue, so it does not really allow any conclusions regarding expression in CAFs. It would be better to perform this analysis in datasets that separate tumor from stromal expression (e.g. Finak G et al. Nature Med. 2008).

Response: As recommended by the reviewer, we have performed the 'normal versus tumour' analysis using the Finak dataset (new Figure 1c). This analysis demonstrates a significant upregulation of *MRC2* (Endo180) expression in the tumour stroma, compared to the normal tissue stroma, along with two other markers of activated CAFs, *ACTA2* (α SMA) and *FAP* (fibroblast activation protein). The analysis of the TCGA dataset (now moved to Supplementary Fig. 1e) has value as it demonstrates that increased expression of *MRC2* in the tumour stroma does not simply reflect a difference in fibroblasts number between the tumour and normal tissue stroma.

1.2. Figure 2,3: The authors tested the functional importance of Endo180 for tumor growth and metastasis, using multiple cell lines and genetic backgrounds, as well as different routes of

inoculation. However, the targeted deletion in Endo180 is not cell type specific, and thus all the host cells in these mice will have the deletion, including immune cells, endothelial cells etc. While there is currently no good specific way to genetically target gene expression in fibroblasts in vivo, the authors should at least acknowledge and discuss this limitation. There is also a way to deal with this experimentally, at least in the primary tumor assays. The authors should substantiate their claim that CAF-derived Endo180 is functionally important by co-injecting orthotopically tumor cells mixed with Endo180^{-/-} fibroblasts (from knockout mice), compared with WT fibroblasts. Currently, the authors' conclusion from these experiments that "Together these data provide strong evidence that upregulated expression of Endo180 in matrix CAFs is required for efficient tumour progression in vivo" is very overstated, considering that the deletion is general, and not specific to matrix CAFs.

Response: We have addressed this major comment in a number of ways

(a) As described in detail in response to Reviewer 2 (comments 2.1/2.2) below we have FACS sorted different cell populations from tumours and demonstrate that Endo180 (*Mrc2*) expression is restricted to CAFs with no detectable expression on D2A1-m2 tumour cells or any other stromal cell type (new Fig. 1b). These data are strengthened by the analysis of publicly available single cell RNA-Seq datasets again showing *Mrc2* expression restricted to subsets of CAF populations (revised Supplementary Fig. 1d). In addition, we have included a new IHC panel (Fig. 2b) of tumours in WT and KO mice illustrating the expression of Endo180 on CAFs, and the lack of Endo180 staining in the KO mice.

(b) We would like to draw the Reviewer's attention to the data showing in Supplementary Fig. 3f,g where we test the hypothesis that the defect in tumour growth in knockout mice is due to loss of *Mrc2* expression in CAFs by inoculating tumour cells into a fibroblast-free tissue i.e., the brain. Using two inoculation routes (direct intracranial and intracardiac) we see no different in tumour growth in the brain. We appreciate that this doesn't provide direct proof that the phenotype is CAF-dependent but it supports our contention.

(c) The referee requested that we perform the experiment of orthotopic co-injecting tumour cells with WT and Endo180^{-/-} CAFs. We agree that this is an obvious experiment to do but there are issues with this approach, which are nicely outlined in a recent "Consensus Statement" piece published in Nature Reviews Cancer by many of the leading CAF experts (Sahai et al., 2020; doi: 10.1038/s41568-019-0238-1). To quote the authors directly

"Two main methods are used to explore CAF functions in vivo: transgenic manipulations and co-injection methods. The latter are simpler to perform as they avoid the need for complex mouse crosses. However, there are some notable caveats. The most challenging is that as tumours grow they will contain a mixture of the co-injected CAFs and fibroblasts derived from the host mouse and, for reasons that are not fully understood, host-derived fibroblasts outgrow co-injected CAFs. In practice, this favours the early evaluation of differences between experimental groups and makes it hard to test longer term phenotypes, such as therapy responses. Transgenic manipulations using Cre-*lox* systems to modulate CAFs overcome these issues but have a different set of issues. The most notable of these is the choice of the Cre driver line. Currently, no CAF-specific Cre driver line exists, and even fibroblast-specific Cre driver lines are complex.

We anticipated that these issues in co-injection studies might be exacerbated when using a syngeneic mouse model as our CAFs were GFP-positive. Nevertheless, we performed the requested experiment injecting mice with D2A1 cells admixed with shNTC or shEndo180 GFP-

positive CAFs. These data are shown in new Supplementary Fig. 3a-d. Similar to the data shown in Fig. 1b, individual D2A1 tumours showed a variability in growth rate. It was notable that co-injection with shNTC CAFs, but not shEndo180 CAFs, led to a more consistent pattern of tumour growth, as monitored by reduced variability in final tumour volume, and an overall modest but not significant increase in tumour size. Immunohistochemical analysis of these tumours readily detected aSMA-positive fibroblasts but negligible numbers of GFP-positive fibroblasts, reinforcing the NRC Consensus Statement that endogenous fibroblasts outgrow co-injected CAFs. In conclusions, due to the limitations of such an approach, the data generated is not conclusive. If the reviewer/editor prefers, these data could be removed from Supplementary Fig. 3

(d) we have included a brief (due to word limit restrictions) statement of the limitations of working with a 'whole body' knockout mouse at the end of the discussion section.

(e) we have toned down our statements concerning the expression of Endo180 in CAFs being required for efficient tumour progression etc - as shown in the marked-up text.

1.3. The title “In vitro assays recapitulate in vivo phenotypes” of the section regarding Figure 4 (p. 7) is not very informative. The in vivo data in figures 2,3 implied a functional role for host-derived Endo180 in facilitating tumor progression. The in vitro data in figure 4 only recapitulates an observation on the effect of fibroblast-expressed Endo180 on tumor cell proliferation in 3D. The other in vitro experiments show that effect is not mediated by secreted factors or by the matrix.

Response: Nature Communications only allow 60 character sub-headings but we appreciate that our original subheading was not optimal. We have changed to 'Reduced contractility and viability of CAFs lacking Endo180'

1.4. Minor comment: there are some typos along the manuscripts that should be fixed.

Response: We apologise for the typographical errors. The revised manuscript has been carefully checked.

Reviewer #2 (Expertise: scRNAseq, Remarks to the Author):

In this manuscript, Jungwirth and colleagues investigate the impact of Endo180-expressing fibroblasts on tumor growth and metastasis. Single-cell and bulk expression profiling of cancer-associated fibroblasts (CAFs) from different tumor types indicate preferential expression of MRC2 (Endo180) on CAFs. As detailed below, there are several major concerns related to the scRNA-seq aspects of this study:

Major comments:

2.1 The in vivo conclusions in this manuscript rely on the assumption that MRC2 (Endo180) expression is specific to fibroblasts. While the immunostaining and publicly available expression data in this manuscript indicate a strong enrichment of MRC2 on fibroblasts, these data, especially the scRNA-seq data, are highly selective and insufficient to make the authors' case. In perusing MRC2 expression across multiple scRNA-seq tumor atlases (<http://tisch.comp-genomics.org/search-gene/?genesearch=MRC2>), MRC2 appears to be variably expressed on malignant cells and myeloid subsets (e.g., macrophages), among other cell types, in various cancers. Whether this expression is translated into productive Endo180

protein products cannot be determined from scRNA-seq data, but at a minimum, these data warrant tempered claims about the specificity of MRC2 for CAFs, particularly in the absence of additional experiments. This is especially critical since the in vivo knockdown experiments in this work presume that the phenotypes observed can be specifically traced to CAFs, but M2-polarized macrophages are also implicated in tumor growth and metastasis in various solid tumors.

2.2 Related to the above, it would be ideal if the authors would sort and profile MRC2 (Endo180) expression on major tumor-associated lineages, including macrophages, in WT and Endo180 KO mice to demonstrate specificity of MRC2 for CAFs within their in vivo system.

Response: We have addressed these Comments 2.1 and 2.2 as follows

(a) As recommended by the referee we inoculated mice orthotopically with D2A1-m2 tumour cells tagged with mCherry. Dissociated tumours were sorted for tumour cells (marked by expression of mCherry and negative for all other markers), macrophages (mCherry-, CD31-, CD45+, F4/80+, PDGFR α -), non-macrophage immune cells (mCherry-, CD31-, CD45+, F4/80-, PDGFR α -), endothelial cells (mCherry-, CD45-, F4/80-, CD31+, PDGFR α +) and CAFs (mCherry-, CD31-, CD45-, F4/80-, PDGFR α +). Population purity was confirmed by RT-qPCR (see new Supplementary Fig. 1b). *Mrc2* (Endo180) expression was only detected in the fibroblast population with no expression on tumour cells, F4/80+ macrophages or any other stromal type. Please note; the referee asked us to perform this in WT and KO mice. In previous publications it has been demonstrated that there is no Endo180 protein detectable in adult Endo180 KO mice (East et al., 2003 EMBO Rep; Engelholm et al., 2003 J Cell Biol). However, to confirm this, we stained 4T1 tumours grown in WT and KO mice and failed to detect any Endo180 protein in the KO tumours (new Fig. 2b).

(b) We are not sure why the reviewer states that scRNA-seq data is highly selective and insufficient. In Fig. 1h we do look at the expression of *Mrc2* in a sc-RNAseq dataset of murine CAFs in order to demonstrate that Endo180 is expressed in only a subset of CAFs. However, in Supplementary Fig. 1d we investigate *Mrc2* expression in a sc-RNAseq dataset of murine melanomas, in which all cell types including the tumour cells are represented. Consistent with the cell type RT-qPCR analysis described above (new Fig 1b) the scRNA-Seq data show that *Mrc2* expression is restricted to the CAF populations showing no expression on lymph node fibroblasts or any of the immune cell types, including macrophages. Please note, at original submission the melanoma dataset had been deposited into BioRxiv. These data have now been published (Davidson et al., 2020 Cell Reports) and as a consequence we have now updated Supplementary Fig. 1d and reference the published manuscript. We also took the opportunity to compare *Mrc2* expression with that of established CAF activation markers α SMA and FAP which further reinforces our conclusion that Endo180 is expressed by a subset of CAFs.

(c) We thank the reviewer for pointing out the Tumor Immune Single-cell Hub (TISCH) datasets. While the collection valuable for studying different immune cells, most datasets unfortunately lack data on fibroblasts. Also, it has to be pointed out that TISCH re-analyses existing data sets. Of the 5 TISCH breast cancer datasets, 2 include only immune cells which do not show any *Mrc2* expression. The other 3 datasets should include both fibroblasts and immune cells and in one case also tumour cells. The latter, a mouse dataset, supports our finding that *Mrc2* is mainly expressed on fibroblasts (see image GSE136206 below).

As mentioned by the reviewer some *Mrc2* expression is also found on malignant T11-Apoec tumour cells. This finding is in accordance with our previous findings that in some cases *Mrc2* can be expressed on a subset of mesenchymal TNBCs (as stated in the introduction, reference 25). Unfortunately, the remaining two datasets are not usable. One dataset shows no expression of *MRC2*

on fibroblasts, myofibroblasts or macrophages. On inspection of the original publication and data, the samples were sorted for tumour cells only and should not include fibroblasts or macrophages, which explains the lack of *MRC2* expression. A similar issue arises with the last dataset which should only include CD45+ve cells. However, the re-analysis by TISCH shows fibroblasts and endothelial cells, with the majority of *Mrc2* expression on fibroblasts (see image below). Again, this implicates a contamination of the samples during the sorting, however, as this is not our own dataset looking into this further would be beyond the scope of our analyses. In conclusion, we have not included any analyses of the TISCH datasets in the revised manuscript as we consider they do not add any additional relevant information to the data presented or referenced in the revised manuscript.

(d) Although our data support our assertion that expression of Endo180 in the tumour stroma is restricted to CAFs, we have included a brief statement at the end of the discussion noting that any strategies to target Endo180 or Endo180-expressing cells would require pre-clinical testing for on-target, off-tumour activity.

(d) The referee correctly states that we, and others, have published that *MRC2* is expressed on some tumour cell types, particularly tumour cells of mesenchymal origin such as subsets of sarcomas (Engelholm et al., 2016 J Pathology) and glioblastomas Huijbers et al., 2010 PLoS One), as well as by a small proportion of highly mesenchymal breast cancers (Wienke et al., 2007 Cancer Research). A statement to this effect was (and remains) included in the Introduction (Page 3).

2.3 The authors provide no methodological details about the scRNA-seq data processing and visualization steps used in Fig. 1h and Supplementary Fig. 1c, making it difficult to properly assess these data.

Response: Regarding Fig. 1h (now Figure 1g). The scRNA-Seq data was downloaded from a published study (Bartoschek et al. 2018 Nat. Commun). The methods section of our manuscript has been updated to read 'scRNA-Seq data of CAFs isolated from MMTV-PyMT tumours was downloaded the published study and analysed as in the original publication. Briefly, single end sequencing reads data was aligned to the mouse genome (assembly: mm10) using STAR aligner (v2.3.0). Aligned reads were quantified and normalised to genes using the Reads Per Kilobase of transcript, per Million mapped reads (RPKM) measurement.'

Regarding Supplementary Fig. 1c (now Supplementary Fig. 1d). At original submission, the data shown in now Supplementary Fig. 1d related to a manuscript deposited on BioRxiv. This manuscript has now been published (Davidson et al., 2020 Cell Reports). In the published manuscript the authors provide a link to their online visualisation software. The Supplementary Fig. 1d legend has now been updated to read 'd tSNE visualisation of colour coded cell types isolated from primary B16-F10 mouse melanoma tumours and draining lymph nodes and subject to single cell RNA-seq (Davidson et al., 2020). Right panels, expression of Mrc2, Acta2 and Fap. Data was visualised using the online tool provided by the authors (<https://melanoma.cellgeni.sanger.ac.uk/>)'.

2.4 The statistical significance of the result in Fig. 1h (now Fig 1g) (right) is missing.

Response: We apologise for this omission. We have now added the statistical significance values (Kruskal Wallis test, $P = 5.6 \times 10^{-50}$).

Minor comments:

2.5 RPKM is not commonly used to represent scRNA-seq expression data. TPM (transcripts per million) or CPM (counts per million) is preferred.

Response: We agree with the reviewer, and could transform RPKM to TPM which is quite straight forward. However, given that we are using a previously processed/published dataset (Bartoschek et al., 2018 Nat Comms), we believe RPKM is an appropriate measure to maintain consistency between the two studies.

2.6 The authors jump from one cancer type (breast, melanoma, colorectal) and species (human/mouse) to another in the bioinformatics analyses on pages 4 and 5 without a clear/compelling rationale or narrative.

Response: We understand this comment and apologise for the lack of a clear narrative. We do feel it is relevant to look at other tumour types as research into the tumour microenvironment should be relevant to other solid tumours. However, to clarify the situation, we now only show data from breast cancers in Figure 1 and moved the analysis of non-breast cancer datasets to the Supplementary Figure 1. In addition, we have endeavoured to clarify the narrative as shown in the marked up text.

Reviewer #3 (Expertise: Matrix remodeling, cancer metastasis, Remarks to the Author):

The manuscript by Jungwirth et al. uses both in vitro and in vivo approaches to dissect the role of stromal cell (fibroblast) Endo180 in the growth of breast cancer cell lines at both primary and secondary metastatic sites. The manuscript is well written, and conclusions are generally supported by the data provided. This work would certainly be of interest to the field and based on the authors addressing my comments below I would recommend publication.

3.1a: The structure of the manuscript appears to jump back and forth a little from in vivo to in vitro and back again. For example, staining for Endo180/ α SMA / Collagen may be better situated with the data on tumor growth in Figure 2 rather than in figure 6, as this would be the logical point to ask the question about what was happening to CAF activation and ECM remodeling in the slower growing tumors.

Response: We agree with the reviewer and had planned to follow their suggestion of moving the tumour data in Fig. 6 to the start of Fig. 3. However, with the inclusion of additional data in Fig. 4f,g and Fig. 5 addressing the Reviewers questions about contraction vs. activation, it makes more sense to address the questions about CAF activation before assessing the stroma composition of the mouse tumours.

Some major questions that remain to be addressed are: Please note - we have combined our responses to Comments 3.1, 3.8 and 3.10.

3.1: Does Endo180 KD alter CAF activation?

3.8: Other than an underpowered (n=2) bulk contraction assay and a cell spreading assay, the manuscript does not present compelling evidence that the differential activation of contractile machinery is dependent on Endo180 expression, and not as a result of differential CAF activation.

3.10: The different stiffnesses of the matrix (F5 and SF6) are known to alter adhesion, fibroblast activation and subsequent contractility. The fact that Endo180 deficient cells remain more spread on a soft matrix does not necessarily confirm they have a contractile dysfunction, if the authors have not ruled out differential activation or adhesion. The authors should consider staining for markers of activation (α SMA) contractility (pMLC, pMYPT) and stress fibers to support their data.

Response: Regarding comments 3.1, 3.8 and 3.10, we would like to thank the reviewer for raising the very interesting question as to whether fibroblast activation is dependent on (or even associated with) increased contractility, which is a hallmark feature of CAFs. First, to clarify, the reviewer comments that the data in Fig. 5a were from cells plated onto soft hydrogels. To note, we had also shown that equivalent data were obtained when cells were plated onto stiff 50 kPa hydrogels or onto glass coverslips (Supplementary Fig. 6a,b). These data, combined with collagen gel contractility data (Fig. 4c,e) strongly suggest that knockdown of Endo180 in fibroblasts results in a reduced cell contractility. However, as the reviewer points out, contractility does not necessarily equate to activation. To address this, we have performed a number of additional experiments:

(a) We have performed immunoblotting and immunofluorescence analysis of our fibroblast populations and demonstrated that (i) there are no notable difference in the levels of the contractile regulators phosphorylated MLC and MYPT, but there is a clear difference in the distribution of pMLC in the shEndo180 fibroblasts where the pMLC is retained more centrally in the cell and does not decorate the actin fibres at the cells periphery (new Fig 5f,g; new Supplementary Fig. 7a), (ii) classic CAF activation markers such as α SMA (*Acta2*) and *Fap*, as well as *Col1a1*, *Col1a2*, and *Myh9* are increased in CAFs compared to normal 3T3 fibroblasts but there is no reduction in these markers in siEndo180 (new Fig. 5) or shEndo180 (Supplementary Fig. 7) CAFs, (ii) using an independent marker of CAF activation i.e., the increased translocation of YAP/TAZ into the nucleus (Calvo et al., 2013 Nature Cell Biology), there is increased YAP/TAZ nuclear translocation in CAFs compared to 3T3 fibroblasts but knockdown of Endo180 has no effect.

In conclusion, these additional data confirm the defect in fibroblast contractility associated with Endo180 deficiency but strongly imply that this is not accompanied by a decrease in CAF activation. As our manuscript goes on to demonstrate, rather this defect in contractility is associated with a loss of fibroblast viability in vivo and in 3D in vitro assays. These findings are now discussed more fully in the Discussion section (see marked up changes to the text).

3.2: Where aSMA is decreased in the tumor histology (F6), is this simply a reflection of less stromal cells overall?

Response: We agree with the referee that a decrease in aSMA staining (Fig. 7a) could reflect that (a) any fibroblasts recruited to the tumour are less activated, (b) there is reduction in the number of fibroblasts in the tumour stroma, and/or as suggested by the referee (c) there are less stromal cells overall. We address (a) versus (b) directly in our manuscript (Figs. 6 and 7) and conclude that a defect in Endo180 expression results in reduced CAF viability. In the revised manuscript we have addressed (c) and provide new data showing no difference in macrophage number in the tumours (new Fig. 7c) and no difference in vascular architecture (new Fig. 7d). In conclusion, we find no evidence that a reduction of aSMA staining in the tumours reflects a of Endo180 results in a global disruption of the tumour stroma.

3.3: Do the D2A1m12 cells have elevated expression of Endo180? It has been previously shown that tumor cells might acquire increased Endo180 expression that plays a role in invasive tumor growth.

Response: The RNA-Seq analysis of D2A1 and D2A1-m12 cells in culture reveals that D2A1-m12 cells show a 5.27 and 2.70 logFC in expression of *Acta2* and Endo180 (*Mrc2*), respectively, consistent with the conclusion that selection of the D2A1-m12 cells results in their acquiring an enhanced ability to remodel and respond to the ECM. These results were validated by RT-qPCR analysis of D2A1 and D2A1-m2 cells directly isolated from primary tumours (new Supplementary Fig. 9c, lower panel). Although it is tempting to conclude that increased *Mrc2* expression alone might drive a more invasive tumour growth, it is important to note that *Acta2* and *Mrc2* expression in D2A1-m12 cells remains substantially lower than the expression levels found in CAFs (new Supplementary Fig. 9c, upper panel). Consequently, it is more likely that the more aggressive phenotype of the D2A1-m12 cells lies in a wider upregulation of the matrisomal genes and pathways highlighted in Fig. 8d-f and Supplementary Fig. 9.

3.4: It would greatly strengthen the manuscript to see overexpression and/or recovery experiments in the low / KD Endo180 cells to support the authors conclusion that this is an Endo180 causal event.

Response: We are confident with the results we present as we obtained strong, highly reproducible data using two different shRNAs to generate stable knockdown lines, two different siRNAs to acutely downregulate Endo180 expression and three different fibroblast lines, as well as using the Endo180^{+/+} and Endo180^{-/-} mice.

3.5: Monoclonal antibodies to Endo180 have been trialed in a number of settings. Do the authors have any evidence that such an approach (or similar Endo180 targeting approaches) would be relevant to this setting, given the title of their manuscript?

Response: As far as we are aware, there is only one report of using monoclonal antibodies to Endo180 therapeutically (Nielsen et al., 2017 Oncotarget). The authors generated an ADC

and demonstrated activity against human Endo180+ U937 tumours in mice. Although encouraging, it is unclear whether their ADC recognises and can kill Endo180+ mouse cells. Consequently, it is not known whether such an approach would be effective for targeting the tumour stroma and, similarly, it is not known whether an ADC would have off-tumour on-target toxicity in normal tissues. We have included a short (due to word limit restrictions) discussion on the therapeutic targeting of Endo180 at the end of the Discussion section.

3.6: The findings that there is less collagen accumulation in Endo180^{-/-} tumors contradicts earlier work where Endo180 knockout led to less tumor burden and increased collagen accumulation in the tumors as a result of defective collagen clearance by tumor-associated stromal cells. Perhaps the authors can comment on this to put their findings into this context.

Response: We apologise that such an explanation was omitted in the original manuscript. In the Curino et al., 2007 JCB manuscript the authors report that crossing Endo180 KO mice with the MMTV-PyMT mouse results in reduced tumour burden and increased accumulation of collagen. The difference in our results likely results from the differences in the models used. PyMT tumours contain few infiltrating fibroblasts and in the Curino manuscript the majority of Endo180-positive cells and collagen accumulation was localised to the stroma surrounding tumour nests. By contrast in the syngeneic models used in our study, we have quantified aSMA-positive cells infiltrating into the tumour mass and intra-tumour fibrillar collagen. From our previous studies, Avgustinova et al., 2016, we show that fibroblasts peripheral to the tumour core have reduced aSMA expression indicating that they are less contractile and therefore, we predict, less susceptible to apoptosis when lacking Endo180 expression resulting in a fibroblast population that is impaired in its ability to remodel the collagen matrix. By contrast, within the tumour core where the fibroblasts are more contractile, lack of Endo180 expression leads to apoptosis and reduced collagen production. The Discussion of the manuscript now includes a section addressing these differing results.

3.7: In F6 (now Fig. 7) the authors show decreased collagen deposition in tumors. Does the KD of Endo180 in CAFs reduce Col1a1/Col1a2 expression in these cells? This would help with my previous comment.

Response: Please see our response above. In brief, KD of Endo180 in CAFs does not alter expression of *Col1a1* or *Col1a2* (new data in Fig.5a) supporting the assertion that Endo180-deficient CAFs show a robust reduction in contractility but not in markers of CAF activation. As clarified in the text, we conclude that the reduction of fibrillar collagens in tumours in Endo180 KO mice results from a loss of CAF viability rather than a reduction in collagen deposition.

3.9: In F5, n=2 plugs is insufficiently powered. Additional repeats should be carried out.

Response: The collagen contraction assays in Figure 5 have been repeated (now n=4). These new data are presented in Fig. 4c and Fig. 4e.

3.11: Given the RNAseq data in F7, the authors should show elevated aSMA in D2A1-m12 cancer cells in vitro in tumor/metastases to support their conclusion. Similarly, some validation of the 9 most significantly enriched collagens / glycoproteins should be undertaken in the in vivo tumors.

Response: As described in our response to Comment 3.3 above we have dissociated D2A1 and D2A1-m12 tumours, isolated the tumour cells and assessed the levels of *Mrc2* (Endo180), *Acta2* (aSMA), and enriched collagens and glycoproteins identified as being upregulated in

D2A1-m12 cells in culture from the RNA-Seq analysis. As a control we included 2 matrisome genes (*Bgn* and *Ltpb2*) that were identified in the RNA-seq analysis as being downregulated in the D2A1-m12 cells. These new data (new Fig. 8 and Supplementary Fig. 9c) validate the *in vitro* RNA-Seq analysis.

3.12: As a supplemental table, it could be helpful to include the table of ECM and ECM-related genes and their changes that the authors say Endo180 is one of. This would allow other researchers to see what may be changing alongside Endo180.

Response: We have included a new supplementary table (new Supplementary Table 2) showing the top 50 mouse matrisome genes (<http://matrisomeproject.mit.edu/other-resources/mouse-matrisome/>) upregulated in D2A1-m12 versus D2A1 cells. Of note, *Mrc2* (Endo180) is not included in the matrisome dataset. As discussed above (comment 3.3) the RNA-Seq analysis shows a 2.70 logFC upregulation of *Mrc2* in D2A1-m12 cells, findings that were validated by RT-qPCR of cultured cells and D2A1-m12 cells isolated from tumours (new Supplementary Fig. 9c). Again, this increased expression is still considerably lower than the levels of *Mrc2* expression in CAFs.

3.13: On this, how do those genes change in the 'omics analysis the authors present in Figure 7. Is the loss of Endo180 leading to a loss of expression of other "matrix CAF" genes that may also explain the phenotypic effects?

Response: The referee's question indeed summarises our conclusion, however there may be some confusion here. Our study focuses on the role of Endo180 in stromal fibroblasts. In Fig. 8 we address how tumour cells might overcome a deficit in stromal Endo180 expression i.e., this is not about loss of Endo180 in the tumour cells but in the stromal fibroblasts. Our conclusion is that in the generation of D2A1-m12 subline we have selected for subclones that can overcome the stromal deficit in the Endo180 knockout mouse by upregulating expression of stromal genes, including Endo180, α SMA and a broad spectrum of 'matrisomal' genes.

3.14: It is not clear why the 3T3 normal fibroblasts were used as the authors state than Endo180 is expressed at a low level in normal stromal fibroblasts.

Response: 3T3 fibroblasts were used in the *in vitro* analysis as 'normal' fibroblasts and although they express *Mrc2* and *Acta2*, and other CAF markers such as *Fap*, this expression is significantly lower than in CAFs (new Fig. 5a; new Supplementary Fig. 9c). Further, 3T3s are less able to contract collagen gels but can be stimulated to do so by TGF β treatment (Fig. 4e).

3.15: Some of the discussion, such as the endosialin KO and MT1-MMP knockout does not seem relevant, unless the authors are attempting to incorporate previous work showing that partial cleavage of collagens by MT-MMP1 leads to increased Endo180 binding and internalization of collagens? However, given the authors case for a non-internalization theory, the focus of the discussion could be tightened. Perhaps instead the authors could discuss their findings in the context of the domain structure of Endo180 and give some insight as to which of the dozen or so domains is likely important in the context of this manuscript especially given that the CTLD domains have been shown to be important in binding glycosylated collagens such as those found in the basement membrane which play an important role in metastasis at both primary and secondary sites.

Response: We agree with the reviewer that the discussion could be 'tightened' (please see marked up text). We have kept in the small section on the phenotype of the endosialin KO

mouse in the Discussion as it illustrates the different functionality of CAF subsets. In the Introduction (Page 3) we describe the key functional domains in Endo180 i.e., that the FNII domain mediates collagen binding, that only the second C-type lectin-like domain (CTLD2) displays Ca²⁺-dependent binding of sugars and that the cytoplasmic domain mediates receptor internalisation into intracellular endosomes. However, this Introductory section and the relevant section of the Discussion has been expanded as the referee's comment highlighted the need to clarify the relationship between collagen binding and receptor internalisation.

REVIEWER COMMENTS

Reviewer #1 (Remarks to the Author):

In the revised manuscript, the authors have addressed all my comments.

Regarding the question whether to include the inconclusive data from co-injection of CAFs (revised Supp. Figure 3): I think the data and explanation should stay. We often omit inconclusive or negative data from manuscripts although they contain valuable information to the community. We can learn from negative data, as long as they are explained, which is the case here.

I now recommend publication and congratulate the authors for an interesting study.

Reviewer #2 (Remarks to the Author):

The authors have done a nice job addressing my critiques from the prior round of review. However, I have a few additional questions that I would like the authors to address prior to publication.

In Fig. 1c, the authors compare expression of MRC2, ACTA2, FAP, and VIM in bulk expression profiles of microdissected stromal cells between breast tumors and adjacent normal tissue. While they find higher levels of MRC2, ACTA2, and FAP in cancer-associated stroma, they identify higher levels of VIM in normal stroma. They then perform a similar analysis on bulk RNA-seq profiles of breast tumors and adjacent normal tissues from TCGA (Supp Fig. 1e). Here, they confirm higher levels of MRC2 in tumor samples and lower levels of VIM in normal adjacent tissue samples. However, ACTA2 (another fibroblast-specific marker) was significantly lower in normal adjacent tissue, in contrast to Fig. 1c. How do the authors reconcile this discrepancy?

To help address this question, perhaps the authors could explore ACTA2, FAP, and VIM expression in the scRNA-seq dataset shown in Fig 1g and VIM expression in the scRNA-seq dataset shown in Supplementary Fig. 1d. Could ACTA2 variably mark MRC2+ cells and VIM show lower expression on MRC2+ cells? For completeness, FAP expression should also be plotted in Supplementary Fig. 1e.

To more definitively establish that “increased MRC2 expression does not simply reflect a proportionate increase in the number of fibroblasts in the tumour samples”, the authors could also explore scRNA-seq data with fibroblasts profiled from both tumor and adjacent normal tissues (e.g., PMID 29988129).

Reviewer #3 (Remarks to the Author):

The authors have made an enormous effort to address the points raised by me and other reviewers during revision, for which they should be commended.

The authors have addressed all of the points of concern and during the revision process greatly improved the manuscript. Whilst it would have been ideal to include the recovery/over-expression experiments included, I accept the authors argument that there is a strong body of evidence using the various KO models to support their conclusions.

Impairment of a distinct cancer-associated fibroblast population limits tumour growth and metastasis

Jungwirth et al. NCOMMS-20-07141B

We would like to thank the three reviewers for their positive responses to our revised manuscript. In this second round of reviews, Reviewer #2 had minor comments that we have been asked to address

Reviewer #2 (Remarks to the Author):

The authors have done a nice job addressing my critiques from the prior round of review. However, I have a few additional questions that I would like the authors to address prior to publication.

In Fig. 1c, the authors compare expression of MRC2, ACTA2, FAP, and VIM in bulk expression profiles of microdissected stromal cells between breast tumors and adjacent normal tissue. While they find higher levels of MRC2, ACTA2, and FAP in cancer-associated stroma, they identify higher levels of VIM in normal stroma. They then perform a similar analysis on bulk RNA-seq profiles of breast tumors and adjacent normal tissues from TCGA (Supp Fig. 1e). Here, they confirm higher levels of MRC2 in tumor samples and lower levels of VIM in normal adjacent tissue samples. However, ACTA2 (another fibroblast-specific marker) was significantly lower in normal adjacent tissue, in contrast to Fig. 1c. How do the authors reconcile this discrepancy?

To help address this question, perhaps the authors could explore ACTA2, FAP, and VIM expression in the scRNA-seq dataset shown in Fig 1g and VIM expression in the scRNA-seq dataset shown in Supplementary Fig. 1d. Could ACTA2 variably mark MRC2+ cells and VIM show lower expression on MRC2+ cells? For completeness, FAP expression should also be plotted in Supplementary Fig. 1e.

To more definitively establish that “increased MRC2 expression does not simply reflect a proportionate increase in the number of fibroblasts in the tumour samples”, the authors could also explore scRNA-seq data with fibroblasts profiled from both tumor and adjacent normal tissues (e.g., PMID 29988129).

Response: We thank the Reviewer for these comments as we agree that in revising our manuscript, we had indeed provided data that required clarification. In particular, we should not have included VIM when analysing bulk tumour stroma or whole tumour datasets as VIM, although expressed by fibroblasts, is also well expressed by a number of other stromal cell types. Consequently, we have

(a) removed the analysis of the TCGA datasets (originally Supplementary Fig. 1e) and the VIM analysis in the Finak et al. dataset (Fig. 1c) and instead directly address the reviewer comments as follows:

(b) included data showing an increased expression of *Mrc2* in two independent CAF cultures isolated from 4T1 mouse mammary tumours compared to normal mouse mammary gland fibroblasts (**new Fig. 1d**).

(c) provided further analysis of the mouse melanoma dataset (Davidson *et al.*) shown in Supplementary Fig. 1d. When quantifying *Mrc2* expression in the normal skin fibroblasts (isolated from non-tumour bearing mice) and the CAF subpopulations in the tumour bearing mice, *Mrc2* expression is significantly higher in the S2 CAFs compared to normal skin fibroblasts (**new Supplementary Fig. 1e**)

(d) as suggested by the reviewer, we performed an equivalent analysis of Lambrechts *et al.* human lung cancer dataset. Again, we see a significantly increased expression of *MRC2* in a subset of tumour-associated fibroblasts in comparison to normal lung fibroblasts (**new Supplementary Fig 1f**).

(e) **revised wording** in the text to state "These data indicate that increased *MRC2* expression in tumours does not solely reflect a proportionate increase in the number of fibroblasts in the tumour stroma"

The reviewer also asked us to explore *Acta2*, *Fap* and *Vim* expression in the Bartoschek *et al.* mouse CAF dataset shown in Fig. 1g. These data are already shown in the Bartoschek manuscript (their Fig. 1e) where *Acta2* is more strongly expressed in the 'vascular CAFs' and 'cycling CAFs', whilst *Vim* and *Fap* show equivalent expression across the 4 CAF subsets. By contrast *Mrc2* (our Fig. 1h) is most strongly expressed in the 'matrix CAFs'.

Finally, the reviewer asks whether *ACTA2* variably marks *MRC2*⁺ cells. None of the datasets examined - Bartoschek, Davidson, Lambrechts - revealed such a correlation, rather all had a combination of *ACTA2*⁺/*MRC2*⁺, *ACTA2*⁻/*MRC2*⁺ and *ACTA2*⁺/*MRC2*⁻ fibroblasts.